# How Much of My Dataset Did You Use? Quantitative Data Usage Inference in Machine Learning

**Yao Tong**,[*] **Jiayuan Ye**,[*] **Sajjad Zarifzadeh, Reza Shokri**
Department of Computer Science, National University of Singapore
{yaotong, jiayuan, s.zarif, reza}@comp.nus.edu.sg

## Abstract

**How much of my data was used to train a machine learning model?** This is a critical question for data owners assessing the risk of unauthorized usage of their data to train models. However, previous work mistakenly treats this as a binary problem—inferring whether *all-or-none* or *any-or-none* of the data was used—which is fragile when faced with real, non-binary data usage risks. To address this, we propose a fine-grained analysis called Dataset Usage Cardinality Inference (DUCI), which estimates the exact proportion of data used. Our algorithm, leveraging debiased membership guesses, matches the performance of the optimal MLE approach (with a maximum error $< 0.1$) but with significantly lower (e.g., $300\times$ less) computational cost.[1]

## 1 Introduction

The increasing legal conflicts over the unauthorized use of datasets to train artificial intelligence models (e.g., the New York Times' lawsuit against OpenAI (The New York Times Company, 2023)) signal the increasingly critical issues about the boundary between intellectual property infringement and fair use, as well as compliance with data privacy regulations (e.g., GDPR (European Parliament and Council of the European Union, 2016) and CCPA (California Legislature, 2018)). This has driven extensive research into data provenance and dataset ownership verification in machine learning (Li et al., 2023; 2020; Tang et al., 2023; Li et al., 2022b; Sun et al., 2022; Sablayrolles et al., 2020; Hu et al., 2022; Zou et al., 2022; Wenger et al., 2022; Maini et al., 2021; Song & Shmatikov, 2019), primarily focusing on implanting backdoor into target models by adding poisoning data to the protected dataset (Wei et al., 2024; Li et al., 2023; 2022b; Tang et al., 2023) or thresholding average statistics over the dataset to conduct hypothesis testing (Maini et al., 2021; Song & Shmatikov, 2019).

However, all these methods are limited to exploring a binary question: whether an entire dataset were used in training a specific model? Such binary methods are observed to be **fragile under partial dataset utilization** (i.e., a practical setting where models are trained on a combination of subsets from multiple data sources). In Figure 1, we plot the original binary prediction given by two representative works (Li et al., 2023; Maini et al., 2021) and the continuous scores we retrieved from their methods (normalized to the range from 0 to 1). (See the implementation details in Appendix B.) While both methods achieve perfect prediction when predicting binary utilization (either none or all), their decision fluctuates in cases of partial usage (regardless of the threshold). For example, both methods will neglect small proportion of usage, and might classify a model as "utilizing" the dataset when 50% of data is included in the training set, yet inferring a model as "not utilizing" the data when usage increases to 60%. Such inconsistencies make these methods unreliable.

In practical applications, we also need to know the extent of dataset usage. For example, according to the Section 107 of the U.S. Copyright Act (United States Code, 1976), determining whether a use qualifies as fair use or copyright infringement must consider the "amount and substantiality of the portion used in relation to the copyrighted work" and "nature of the copyrighted work" (i.e., different

---

[*]Co-first author
[1]Code is at `https://github.com/privacytrustlab/ml_privacy_meter/tree/master/research`.

use cases require different thresholds). However, none of the existing methods can be extended in a straightforward way to predict the proportion of dataset usage, due to their specific assumptions and objectives. Specifically, our goal is to identify a suitable continuous score and inferring dataset usage *without any modifications* to the data itself. This later constraint is critical for broadening the applicability of our methods to scenarios where *all* prior *binary* watermarking approaches (Guo et al., 2023; Li et al., 2023; Tang et al., 2023; Hu et al., 2022; Li et al., 2022b; Wenger et al., 2022; Sablayrolles et al., 2020) fall short: 1) scenarios where altering high-quality data is undesirable or where data integrity must be preserved (e.g., in healthcare applications); 2) situations involving previously released datasets that have likely already been incorporated in trained models.

***Can we reliably and cost-effectively infer the proportion of the target dataset used in a model?*** We introduce the Dataset Usage Cardinality Inference (DUCI) problem in machine learning, via a paradigm shift from binary formulations to a granular assessment of dataset utilization, and propose an efficient and accurate algorithm to precisely estimate the proportion of a dataset used in model training. Specifically,

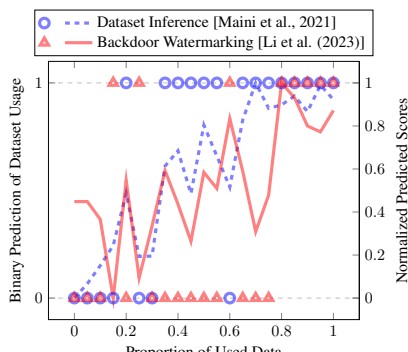

1. We formally define the DUCI problem and its challenge from unavoidable membership guesses errors (Section 3).
2. We develop a novel debiasing procedure that is **applicable to all membership identification methods** (equation 4).
3. We design an unbiased estimator for data usage by debiasing individual membership guesses (Section 4) and quantify uncertainty in DUCI by constructing confidence intervals via Lyapunov Theorem (Billingsley, 2017) and validate them through extensive experiments (Figure 3).

Figure 1: Prior (binary) data inference methods are unreliable and fluctuate between 0 and 1.

4. Our experiments confirm the accuracy and cost-effectiveness of our method (Section 5.3): it provides reliable proportion estimation **maximum error** $< 0.1$ with as few as **one** reference model (Figure 2), and requires $630\times$ less computational budget than idealized methods while maintaining comparable performance (Table 1).
5. Our debiasing method can be extended to group-level (Table 2), and effectively address the challenge of practical quality-dependent dataset sampling schemes.
6. Our methods can also be applied to language models for practical tasks such as quantifying book copyright infringement (Table 4).

## 2 PRELIMINARIES

**Notations**. We use $\mathcal{T}$ to denote the training algorithm that, in addition to the model architecture, may have access to auxiliary public information, e.g., a population data pool $\rho$. We consider the standard supervised learning setting, where a trained model $f_\theta$ processes an input $x$ to produce a prediction $f_\theta(x)$ (e.g., the predicted probability on target class in image classification or the inverse perplexity in text generation). Membership $m_i$ of a data point $x_i$ for the training set $D$ of a model $f_\theta$ is $m_i = 1$ if $x_i \in D$ (a member), and $m_i = 0$ if $x_i \notin D$ (a non-member). We define the term *membership guess* $\hat{m}_i$ as the predicted membership status of a data point $x_i$ by a membership inference algorithm.

### 2.1 MEMBERSHIP INFERENCE (MI)

MI test whether a data point $x$ is part of the (unknown) training set of a target model $f_\theta$. We briefly present the state-of-the-art MI algorithm that enables the best performance in our experiments.[2]

**Robust Membership Inference Attack (RMIA) (Zarifzadeh et al., 2024)**. The method combines multiple likelihood ratio tests, each testing whether the model output $f_\theta$ is more likely when $x$ is in the training set than when it is replaced by a random data point $z$ from the population (i.e., $x$ is not in the training set).

$$\hat{m} \equiv \text{MIA}(x; \theta) = \mathbb{1}_{[P_{z \sim \rho}(LR_\theta(x,z) > 1) \geq \beta(x)]} \text{ for threshold } \beta(x), \quad (1)$$

---

[2]We present the results of other MI algorithms in Appendix H.4 for comparison.

where $\rho$ is the population distribution, and $LR_\theta(x, z) = \frac{P(\theta|x)}{P(\theta|z)}$ is the pairwise likelihood ratio between $x$ and $z$, which can be computed as $(P(x|\theta)/P(x))(P(z|\theta)/P(z))^{-1}$. Here, $P(x|\theta) = f_\theta(x)$ and $P(x)$ is a normalization constant for $x$ calculated by integrating over some reference models trained with the same structure and training data distribution (but not the same training dataset) as the target model $f_\theta$. Additional discussions of related works are deferred to Appendix A.

# 3 DATASET USAGE CARDINALITY INFERENCE PROBLEM

In this section, we formulate the problem of Dataset Usage Cardinality Inference (DUCI) in machine learning and discuss the challenges of solving this problem.

## 3.1 DATASET USAGE AS A NON-BINARY INFERENCE PROBLEM

**Problem formulation**. Given a model $\theta$ and a target dataset $X$, our objective is to infer what fraction of data points in $X$ is used in the training of $\theta$.

The model $\theta$ is trained on a proportion $p = \frac{1}{|X|} \sum_{i=1}^{|X|} (m_i := \mathbf{1}_{x_i \text{ selected}})$ of the dataset $X$, using a training algorithm $\mathcal{T}$. Since the training set may include data from other sources, we model this by introducing a population pool within $\mathcal{T}$, from which additional data can be sampled. The Dataset Usage Cardinality Inference (DUCI) algorithm $\mathcal{A}$—acting as an agent for the dataset owner with full access to $X$—aims to estimate (denoted as $\hat{p}$) the overall usage proportion $p$, given black-box access to $\theta$ and knowledge of the training algorithm $\mathcal{T}$.

In this work, we assume each data point $x_i$ in $X$ is independently included in the training set of $\theta$ with probability $\gamma_i$. Let the vector $\boldsymbol{\alpha} = [\gamma_i]_{i=1}^{|X|}$ be the sampling probabilities for all records in $X$. The pipeline is as follows:

$$\text{Dataset } X = [x_i]_{i=1}^{|X|} \qquad\qquad X$$
$$\text{Independently include } x_i \text{ with probability } \gamma_i \downarrow \qquad\qquad \downarrow$$
$$\text{Selected } p \text{ proportion of } X \xrightarrow{\text{Training Algorithm } \mathcal{T}} \theta \xrightarrow{\text{Black-Box}} \mathcal{A} \to \hat{p} \qquad (2)$$

We aim to design a DUCI algorithm $\mathcal{A}$ that, for any possible dataset usage proportion value, achieves a low prediction gap. That is, minimizing the maximum mean absolute error[3] across usage proportions, where the mean ensures robustness to algorithmic randomness, *i.e.*,

$$\min_{\mathcal{A}} \max_{\gamma_1, \cdots, \gamma_{|X|} \in [0,1]} \mathbb{E}\left[|p - \hat{p}|\right], \qquad (3)$$

where $p = \frac{1}{|X|} \sum_{i=1}^{|X|} m_i$ denotes the ground-truth proportion, $\hat{p} = \mathcal{A}(\theta, X)$ denotes the estimated proportion, and the expectation is over random trials of the DUCI pipeline (Equation (2)), i.e., over data sampling $m_i \sim \text{Bernoulli}(\gamma), i = 1, \ldots, |X|$ and the random coins of the training algorithm.

## 3.2 CHALLENGES IN DATASET USAGE CARDINALITY INFERENCE

A natural strategy for inferring dataset usage is to sum the membership inference guesses over individual training data. However, we observe that this method suffers from poor performance (Table 1) due to the inherent errors in inferring membership of individual data points.

**Errors in optimal point-wise membership inference:** Membership inference at the level of individual training data has a high error when there are inherent randomness in the training pipelines (Ye et al., 2022) (such as data sampling and randomness of the training algorithm). When the training algorithm satisfies certain constraints in terms of output indistinguishability, one can prove that even *the theoretically most informed and optimal membership estimate* ($\hat{m}_i$) has an error that increases with the level of indistinguishability (Kairouz et al., 2015; Steinke et al., 2024). In practice, the error of existing membership inference methods is not only due to these inherent errors but also due to their inability to extract all membership information from model outputs.

---

[3]See Appendix D and discussions on Figure 3 for why additive error is a more appropriate measure and why a multiplicative guarantee for the DUCI problem is impossible.

**Challenge: biases arising from the accumulation of errors:** The aforementioned per-point membership identification errors accumulate and introduce a bias in solving the DUCI task. This bias is clearly illustrated in Table 1. In the next section, we will discuss strategies to mitigate this bias, and show that aggregating debiased membership guesses can serve as a promising proportion estimator.

# 4 UNBIASED DATASET USAGE INFERENCE FROM AGGREGATION OF INDIVIDUAL GUESSES

**Given the unavoidable errors in membership identification methods, how can we achieve a reliable (e.g., unbiased) estimation of the proportion?**

Denote $X$ as the dataset of interest, and $\mathcal{M}$ as the distribution of the ground-truth membership vector. We have $\mathcal{M} = \text{Bernoulli}(\gamma_1) \times \cdots \times \text{Bernoulli}(\gamma_{|X|})$, i.e., the probability of each data record $x_i$ being used is $\gamma_i$ (Section 3.1). For each trial, a target model $\theta$ is trained by randomly drawing a membership vector $\mathbf{m} = (m_1, \cdots, m_{|X|}) \sim \mathcal{M}$, and including the $i$-th data record of $X$ (denoted as $x_i$) if $m_i = 1$ for each $i$. As the proportion $p(\mathbf{m}) = \frac{1}{|X|} \sum_{i=1}^{|X|} m_i$ is a value that varies across trials due to the randomness in $\mathbf{m}$, our goal is to construct a reliable estimator $\hat{p}(\mathbf{m})$ that is accurate (in expectation) *across* random trials. To do so, we design an estimator $\hat{p}(\mathbf{m})$ that satisfies $\mathbb{E}_{\mathbf{m} \sim \mathcal{M}}[\hat{p}(\mathbf{m})] = \mathbb{E}_{\mathbf{m} \sim \mathcal{M}}[p(\mathbf{m})]$, i.e., is unbiased over trials. For brevity, when the context is clear, we use $p$ and $\hat{p}$ to refer to $p(\mathbf{m})$ and $\hat{p}(\mathbf{m})$, respectively.

**Construct $\hat{p}$ from debiased individual MIA guesses.** By definition, the expected inclusion proportion $p$ is the sum of inclusion probability for each sample . Thus, by averaging a per-sample estimate for the inclusion probability of each data record, we design the below estimator:

$$\hat{p} = \frac{1}{|X|} \sum_{i=1}^{|X|} \hat{p}_i \quad \text{where } \hat{p}_i = \frac{\hat{m}_i - \Pr(\hat{m}_i = 1 | m_i = 0)}{\Pr(\hat{m}_i = 1 | m_i = 1) - \Pr(\hat{m}_i = 1 | m_i = 0)}. \tag{4}$$

Here, $\hat{p}_i$ estimates the inclusion probability $\gamma_i$ for data record $x_i$; $\hat{m}_i \in \{0, 1\}$ is the membership guess (given by an arbitrary membership inference method) for data record $x_i$ (with ground-truth membership $m_i \in \{0, 1\}$); and $\Pr(\hat{m}_i = 1 | m_i = 0)$ and $\Pr(\hat{m}_i = 1 | m_i = 1)$ are the per-record False Positive Rate (FPR) and True Positive Rate (TPR) for inferring the membership of each $x_i \in X$ (constants across trials).

We refer to the computation of $\hat{p}_i$ as **debiasing process** and assume TPR $\neq$ FPR, which is reasonable given that membership identification algorithms are designed to discriminate between member and non-member data points, not to act randomly. In our experiment, we use the state-of-the-art MIA methods to compute the membership guesses $\hat{m}_i = \text{MIA}(x_i; \theta)$ (as introduced in Section 2), but note that our approach is *flexible and applicable to any membership prediction technique*.

**Unbiasedness of $\hat{p}$ and its intuition.** How can an estimator $\hat{p}$ be unbiased across trials? That is, and estimator $\hat{p}$ satisfying $\mathbb{E}_{\mathbf{m} \sim \mathcal{M}}[\hat{p}(\mathbf{m})] = \mathbb{E}_{\mathbf{m} \sim \mathcal{M}}[p(\mathbf{m})]$. Note that as $m_i \sim \text{Bernoulli}(\gamma_i)$, $\mathbb{E}[m_i] = \gamma_i$ for any $i = 1, 2, \ldots, |X|$ over the random trials. By the additivity of expectation, we have $\mathbb{E}[p] = \mathbb{E}[\frac{1}{|X|} \sum_{i=1}^{|X|} m_i] = \frac{1}{|X|} \sum_{i=1}^{|X|} \gamma_i$. Thus, estimating $\gamma_i$ and aggregating across all data points serves as an ideal solution. Since $\gamma_i = \Pr(m_i = 1) = 1 - \Pr(m_i = 0)$, the rule of total probability ensures the following relationship:

$$\mathbb{E}[\hat{m}_i] = 1 \cdot \Pr(\hat{m}_i = 1) + 0 \cdot \Pr(\hat{m}_i = 0)$$
$$= \Pr(\hat{m}_i = 1 | m_i = 0) \cdot \Pr(m_i = 0) + \Pr(\hat{m}_i = 1 | m_i = 1) \cdot \Pr(m_i = 1) \tag{5}$$

Rearranging terms in Equation (5) yields $\hat{p}_i$ in Equation (4). Consequently,

$$\mathbb{E}[\hat{p}] = \mathbb{E}\left[\frac{1}{|X|} \sum_{i=1}^{|X|} \hat{p}_i\right] = \frac{1}{|X|} \sum_{i=1}^{|X|} \frac{\mathbb{E}[\hat{m}_i] - \Pr(\hat{m}_i = 1 | m_i = 0)}{\Pr(\hat{m}_i = 1 | m_i = 1) - \Pr(\hat{m}_i = 1 | m_i = 0)},$$

which simplifies to $\mathbb{E}[\hat{p}] = \frac{1}{|X|} \sum_{i=1}^{|X|} \gamma_i = \mathbb{E}[p]$. Therefore, $\hat{p}$ is an **unbiased estimator** of the expected inclusion proportion. We next discuss how to compute the TPR and FPR in Equation (4).

**Efficient and accurate estimation of TPR and FPR.** To compute the estimators in Equation (4), we require the probabilities $\Pr(\hat{m}_i = 1 | m_i = 0)$ (FPR$_i$) and $\Pr(\hat{m}_i = 1 | m_i = 1)$ (TPR$_i$). A direct, but

computationally expensive, approach is to estimate these probabilities using $N$ empirically trained reference models, where the training datasets are known. However, this method incurs a sampling error of order $\Omega\left(\frac{1}{\sqrt{N}}\right)$ (i.e., the approximation error of $\text{TPR}_i$ and $\text{FPR}_i$ can be notable when $N$ is small). To reduce error and computational cost, we propose estimating TPR and FPR across the entire dataset $X$, which requires as few as a **single** reference model and can **reuse** models from the MIA process. This simplification is justified because the proportion $p$ is a dataset-level statistic, i.e., $\mathbb{E}[p] = \frac{1}{|X|}\sum_{i=1}^{|X|}\gamma_i$.[4]Denote $\theta_1,\dots,\theta_N$ as $N$ reference models (where $N$ can be 1), each trained on randomly sampled *halves* of the dataset $X$, we can compute:

$$\text{FPR} = \frac{1}{|X|}\sum_i \text{FPR}_i \approx 2\sum_{j=1}^{N}\sum_{i=1}^{|X|}\frac{\mathbf{1}_{MIA(\theta_j,x_i)=1 \text{ and } x_i \notin \theta_j}}{N \cdot |X|} \tag{6}$$

$$\text{TPR} = \frac{1}{|X|}\sum_i \text{TPR}_i \approx 2\sum_{j=1}^{N}\sum_{i=1}^{|X|}\frac{\mathbf{1}_{MIA(\theta_j,x_i)=1 \text{ and } x_i \in \theta_j}}{N \cdot |X|} \tag{7}$$

The confidence interval analysis below reveals that this empirical average is only subject to $O(\frac{1}{\sqrt{N \cdot |X|}})$ sampling error (i.e., the standard deviation of the empirical estimation of FPR and TPR over the randomness of $\theta_j$ and MIA algorithm). See Appendix G for the proofs. Thus, employing larger datasets and more reference models enhances the accuracy of our debiasing method.

**Confidence interval and uncertainty estimates for Dataset Usage Cardinality Inference** Modern machine learning algorithms and their data sampling processes are highly randomized. To capture the intrinsic uncertainty and confidence in data usage inference, we resort to the Lyapunov Central Limit Theorem (CLT) (Billingsley, 2017) to compute confidence intervals for our estimation method via aggregate statistics. Specifically, if the individual statistics $\hat{p}_1,\cdots,\hat{p}_{|X|}$ in Equation (4) are independent [5] and the Lyapunov condition is satisfied (see the proofs in Appendix F.2), then their aggregation converge to a Gaussian distribution as $|X|$ increases. By applying Lyapunov CLT and using the sample variance to estimate the average variance (see details in Appendix F), we obtain the following approximate $95\%$ confidence interval for estimating $\mathbb{E}[p]$.

$$\hat{p} - t_{\alpha/2}\sqrt{\frac{s^2}{|X|}} \leq \mathbb{E}[p] \leq \hat{p} + t_{\alpha/2}\sqrt{\frac{s^2}{|X|}} \tag{8}$$

Here $\hat{p} = \frac{1}{|X|}\sum_{i=1}^{|X|}\hat{p}_i$, $\alpha = 0.05$, $t_{\alpha/2}$ is the $t$ critical value, and $s^2 = \frac{1}{|X|-1}\sum_{i=1}^{|X|}(\hat{p}_i - \hat{p})^2$.

## 5 EXPERIMENTAL SETUP AND RESULTS

### 5.1 EVALUATION SETTINGS

We considered the general uniformly random sampling setting and special data selection setting, ensuring that $p$ is controllable over the random trials (as described in equation 3). The comparison experiments in Section 5.3 are conducted under the uniformly random sampling, and Section 5.4 extends the discussion to the special sampling scenarios. For uniformly random sampling, we examine all methods under dataset usage proportion $p \in [0,1]$ with a granularity of $5\%$, i.e., $p = 0.00, 0.05, \cdots, 1.00$. For each $p$, we evaluate the (mean absolute) error $\mathbb{E}[|p - \hat{p}|]$ of proportion estimation over 30 random trials. In each evaluation trial, we train the target model $\theta$ on freshly sampled random $p$ proportion of the protected dataset $X$, combined with randomly sampled remaining data from a population pool (that does not overlap with $X$). For settings with specialized data sampling, we constructed the data selection probability vector $\mathbf{p}$ for each ground-truth $p$ using data

---

[4]To clarify further, under dataset-level estimates, we have $\frac{1}{|X|}\sum_i \mathbb{E}[\hat{p}_i] = \mathbb{E}[p] + \frac{\text{Corr}_i(\text{TPR}_i - \text{FPR}_i, \gamma_i)}{\text{TPR} - \text{FPR}}$ (See derivation in Appendix E). This suggests that for many practical samplings (e.g., i.i.d. sampling), this simplification yield an unbiased estimator for $p$. For special sampling methods, we can debias subgroups to achieve the unbiasedness (as discussed in Table 2).

[5]We experimentally validate that $\hat{p}_1,\cdots,\hat{p}_{|X|}$ are approximately independent (Figure 4) and use $> 30$ individual observations to compute valid confidence intervals via Lyapunov CLT.

selection strategies (Paul et al., 2021) and sample data from $X$ for model training accordingly. We then compute the mean absolute error (MAE) in the same way.

Considering the evaluation cost of training target models on various proportions $p$, we mainly assess all methods in Section 5.3 across three model architectures: a standard 5-layer fully connected network (FC-5), a wide ResNet with width 2 (WRN28-2) (Zagoruyko & Komodakis, 2016), and ResNet-34 (He et al., 2016), as well as three benchmark datasets: CIFAR-10 (Krizhevsky et al., 2009), CIFAR-100 (Krizhevsky et al., 2009), and Tiny-ImageNet (Le & Yang, 2015). We further validated the applicability of our methods to language models through a book copyright infringement case study on GPT-2 (Radford et al., 2019), using the BookMIA dataset (Shi et al., 2023). To implement our unbiased proportion estimator, we use MIA to predict the membership guesses, i.e., $\hat{m}_1, \cdots, \hat{m}_{|X|}$. We report the best-performing MIA (i.e., RMIA in Section 2.1) and defer discussions on other MIAs to Appendix C and Appendix H.4. See Algorithm 1 for a pseudocode of our method and see Appendix H for more implementation details.

## 5.2 BASELINES

**Vanilla Membership Inference Attacks (MIA) baselines** To demonstrate the importance of debiasing, we compare our method against two intuitive baselines: direct averaging of binary MIA guesses and real-valued MIA scores outputted by existing MIA methods (Zarifzadeh et al., 2024; Carlini et al., 2022). We assign $N$ reference models with each trained on half of dataset $X$ to MIA baselines:

- **MIA Guess:** This involves directly counting MIA membership predictions, where the proportion is calculated as the fraction of member predictions: $\hat{p} = \frac{1}{|X|} \sum_{i=1}^{|X|} \hat{m}_i$. For a fair comparison, we use the same MIA algorithms that were employed in our method.
- **MIA Score:** This takes the average of MIA confidence scores across all data points: $\hat{p} = \frac{1}{|X|} \sum_{i=1}^{|X|} \hat{c}_i$, where $\hat{c}_i$ is the confidence score of RMIA (i.e., probability before thresholding).

**Maximum Likelihood Estimate (MLE) baselines** When computation cost is not a concern, the below computationally expensive MLE estimators can serve as an idealized baseline:

$$\hat{p}_{MLE}(o) = \arg \max_q \Pr(o|p = q), \tag{9}$$

where $\Pr(o|p = q)$ is the conditional probability of observations $o$ (generated from the trained model $\theta$) given that $q$ proportion of the records in the target dataset $X$ is used in training. This probability $\Pr(o|p = q)$ can be empirically approximated via training many ($N$) reference models on freshly sampled random $q$-fraction of the target dataset $X$ for each of 21 possible proportion value $q$, employing the same algorithm $\mathcal{T}$ that was used for training the target models (in total $21N$ reference models). Regarding the observation $o$ in the MLE baselines, we consider two choices as follows.

- **Joint logits (Joint-Logit) of all records in target dataset (high-dimensional):** Building on prior research (Carlini et al., 2022) which demonstrated that scaled logits for individual data points can serve as effective observations and empirically follow a Gaussian distribution, we naturally extend this approach to a dataset-wide scale. We concatenate these point-wise logits into high-dimensional joint logits and model their distribution using a multivariate Gaussian.

$$\Pr(o|p = q) = \Pr(\text{logit}(X; \theta)|\mathcal{N}(\boldsymbol{\mu_q}, \boldsymbol{\Sigma_q})), \tag{10}$$

where $\text{logit}(X; \theta)$ is the the logits of model $\theta$ on all records in $X$, and $\mu_q$ and $\boldsymbol{\Sigma_q}$ are its empirical mean and covariance matrix across $N$ reference models trained with $q$ proportion of $X$.

- **Averaged logits (Avg-Logit) of all records in the target dataset (one-dimensional):** To enable more accurate empirical mean and covariance estimation, we reduce observation dimensionality by using average statistics (i.e., the average of logits) over the dataset $X$, akin to Maini et al. (2021). The likelihood is then modeled as one-dimensional Gaussian.

$$\Pr(o|p = q) = P\left(\frac{1}{|X|} \sum_{x \in X} \text{logit}(x; \theta) \Big| \mathcal{N}(\mu_q, \sigma_q^2)\right),$$

where $\mu_q$ and $\sigma_q$ are the empirical mean and variance of the averaged logits of all records in $X$ across $N$ reference models trained with $q$ proportion of $X$.

Table 1: Maximum mean absolute error (MAE) ($\max_{p \in [0,1]} \mathbb{E}[|p - \hat{p}|]$) over 21 proportions $p$, each derived from 30 trials, of all methods across various datasets and model architectures within the same computational budget (# reference models). The lowest error in each test is highlighted in bold. Grey rows are the idealized computationally expensive MLE baselines.

| #Ref | Methods | CIFAR-10 | | CIFAR-100 | | Tiny-ImageNet | |
|---|---|---|---|---|---|---|---|
| | | FC-5 | WRN28-2 | ResNet34 | WRN28-2 | ResNet34 | WRN28-2 |
| 1 | MIA Guess | 0.3291 | 0.4819 | 0.1611 | 0.1676 | 0.0889 | 0.1478 |
| | MIA Score | 0.3619 | 0.4486 | 0.2950 | 0.3220 | 0.2909 | 0.3040 |
| | **Ours** | **0.0678** | **0.1836** | **0.0534** | **0.0873** | **0.0879** | **0.0580** |
| 2 | MIA Guess | 0.2617 | 0.4801 | 0.1637 | 0.1458 | 0.0859 | 0.1131 |
| | MIA Score | 0.3587 | 0.3865 | 0.2687 | 0.3074 | 0.2822 | 0.2952 |
| | **Ours** | **0.0380** | **0.1787** | **0.0421** | **0.0851** | **0.0723** | **0.0463** |
| 4 | MIA Guess | 0.2797 | 0.4350 | 0.1468 | 0.1401 | 0.0746 | 0.0889 |
| | MIA Score | 0.3571 | 0.3822 | 0.2571 | 0.2836 | 0.2593 | 0.2739 |
| | **Ours** | **0.0345** | **0.1634** | **0.0325** | **0.0774** | **0.0564** | **0.0242** |
| 8 | MIA Guess | 0.2831 | 0.3862 | 0.1493 | 0.119 | 0.0768 | 0.1099 |
| | MIA Score | 0.3022 | 0.3621 | 0.2552 | 0.2824 | 0.2588 | 0.2759 |
| | **Ours** | **0.0296** | **0.1127** | **0.0180** | **0.0554** | **0.0368** | **0.0178** |
| 42 | MLE (Joint-Logit) | 0.2450 | 0.2230 | 0.1983 | 0.2667 | 0.2050 | 0.2067 |
| | MLE (Avg-Logit) | 0.0533 | 0.1383 | 0.0567 | 0.0617 | 0.0383 | 0.0583 |
| | MIA Guess | 0.2516 | 0.2889 | 0.086 | 0.0797 | 0.0377 | 0.0867 |
| | MIA Score | 0.2948 | 0.3692 | 0.2530 | 0.2920 | 0.2576 | 0.2867 |
| | **Ours** | **0.0271** | **0.0722** | **0.015** | **0.0339** | **0.0124** | **0.0179** |
| 630 | MLE (Joint-Logit) | 0.1400 | - | - | - | - | - |
| | MLE (Avg-Logit) | 0.0283 | - | - | - | - | - |

## 5.3 MAIN RESULTS

In this section, we compare the performance of our method (Algorithm 1) with the baselines in Section 5.2 in terms of estimation quality and efficiency.

**Comparison under the same computation budget** Table 1 shows an overview of the comparison results under different datasets and model architectures while keeping the total computational budget fixed. Note that for our debiasing process, single reference model is enough and those reference models for launching MIA can be reused. For the MIA Guess baseline, we report the best performance achieved across all candidate MIA methods. We first observe that our method consistently outperform all baselines under all settings under 42 reference models (which is the minimal number of reference models required for MLE baselines), whether on well-generalized setting such as WRN28-2 on CIFAR-10 or more challenging datasets like CIFAR-100 and Tiny-Imagenet. These latter datasets have fewer data records per class, thereby increasing their susceptibility to memorization and generally making them easier tasks for DUCI. We then tested the case with extremely limited computational budget (i.e., 1-8 reference models). This is challenging, as the scarcity of empirical samples for estimating FPR and TPR will lead to less accurate debiasing. Additionally, existing MIAs themselves also suffer from reduced strength when the number of reference models is small (Carlini et al., 2022; Zarifzadeh et al., 2024). Remarkably, our method achieves significantly smaller estimation error than vanilla MIA baselines even with as few as 1 model, i.e., *our method can provide reliable proportion estimates even under a severely restricted computational budget.*

Figure 2 further shows that, for all methods, prediction error generally decreases as the number of reference models increases, under the setting of FC-5 trained on CIFAR-10. This improvement is expected, as more reference models provide more accurate estimation of the likelihood $\Pr(o|p)$ for MLE methods, the distribution for the MIAs, and TPR/FPR for our debiasing methods (Section 4), thereby reducing prediction errors. Notably, our method outperforms MLE methods more under less computation budgets This is because the accuracy of MLE methods relies heavily on the quality of approximating the $\Pr(o|p)$ for all possible $p \in [0, 1]$, a task that becomes challenging with a limited

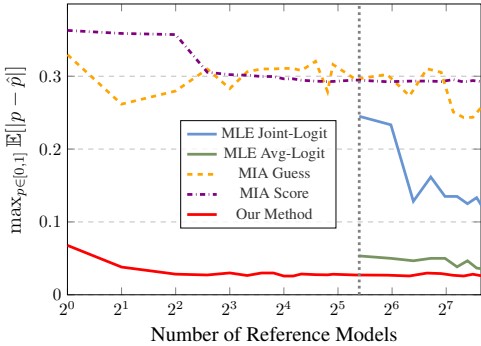

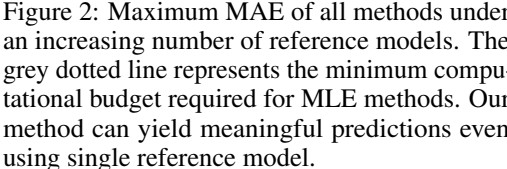

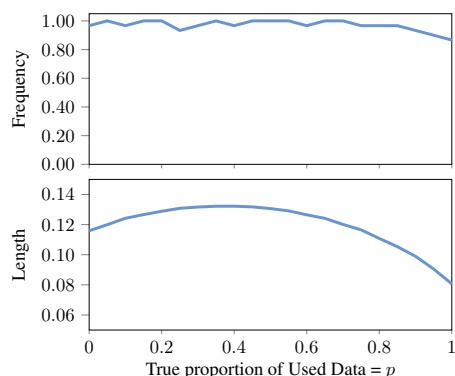

Figure 2: Maximum MAE of all methods under an increasing number of reference models. The grey dotted line represents the minimum computational budget required for MLE methods. Our method can yield meaningful predictions even using single reference model.

Figure 3: *(Confidence Interval)* **Top:** Frequency of the true proportion $p$ falling within the predicted 95% confidence interval for different $p$. The frequency was computed across 30 interval predictions for each $p$. **Bottom:** Average width of the predicted interval over 30 tests for each $p$.

computation budget. Our results indicate that, under such constraints, the distribution estimation process required for MLE introduces more significant bias compared to our method.

We also allocate a substantial budget—a total of $30 \times 21 = 630$ reference models—to the strongest baseline, the MLE methods (grey rows in Table 1), to fully exploit their potential. Note that MLE methods, given extra information on candidate proportions (i.e., hypotheses) and accurate distribution estimation, can serve as idealized baselines according to Neyman & Pearson (1933). The comparison demonstrates that our method can achieve *comparable estimation quality* to MLE methods while incurring *significantly lower computational costs*, i.e., $100\times$ smaller. Due to the huge computational costs, we only presents the results of FC-5 models trained on CIFAR-10 for this comparison.

**Confidence interval for dataset usage inference** Our methodology also aims to capture the uncertainty inherent in dataset usage inference. To evaluate the efficacy of our interval estimation, we conducted tests on the same series of target models—30 trials for each proportion—employing a significance level of $\alpha = 0.05$. The confidence intervals were computed in accordance with Equation (22), where the degrees of freedom is $|X| - 1 = 499$, leading to a t-value of $t_{\alpha/2} = 1.96$.

Figure 3 (Top) displays the frequency of the true proportion $p$ falling within the predicted 95% confidence interval. There is a consistently high frequency (over 0.9), which demonstrates the high coverage of our interval predictions. Moreover, the intervals generally have small length, as shown in Figure 3 (Bottom), suggesting that the overall uncertainty of our proportion prediction is small (around 0.1). Note that here we derive the additive confidence interval (absolute uncertainty) instead of the multiplicative confidence interval (relative uncertainty) because the error in Dataset Usage Inference task is not continuous. (e.g., For a small dataset in size 100, the unit of error is 1%.) Additionally, we observe that our confidence intervals exhibit greater length when $p = 0.5$. This is reasonable as the number of possible $p$-proportion subsets of the target dataset is the largest when $p = 0.5$, thus making dataset usage inference inherently more complex at $p = 0.5$.

## 5.4 DISCUSSIONS

We next discuss the properties, further improvement and potential application of our method.

**Dataset usage inference under special sampling challenges.** While our dataset-level debiasing methods (Equation (6) and Equation (7)) are statistically guaranteed to be unbiased under common

Table 2: Relative stability of our methods under special sampling, measured by Maximum mean absolute error (MAE) ($\max_{p \in [0,1]} \mathbb{E}[|p - \hat{p}|]$) over 21 proportions.

| Sampling | MLE (Best) | MIA Guess | Ours |
|---|---|---|---|
| Random | 0.0533 | 0.2933 | **0.0619** |
| Significant First | 0.6117 | 0.3876 | **0.1089** |
| Insignificant First | 0.6301 | 0.3787 | **0.1077** |

Table 3: Dataset usage estimation error decreases as the (relative) size of the target dataset increases for all methods. Errors are measured by Maximum Mean Absolute Error (MAE) $\left(\max_{p \in [0,1]} \mathbb{E}[|p - \hat{p}|]\right)$ across 21 proportions using WRN28-2 trained on CIFAR-10

| $|X|$ | MLE (Best) | MIA Guess | Ours |
|---|---|---|---|
| 500 | 0.1383 | 0.2889 | **0.0722** |
| 5000 | 0.0849 | 0.268 | **0.0528** |
| 25000 | 0.08 | 0.2659 | **0.0442** |

sampling scenarios like i.i.d. and uniformly random sampling (see Footnote 4 for clarification), the question remains: ***can they also achieve near-unbiasedness under special or adversarial sampling?*** To the best of our knowledge, all previous work considers a uniformly random (or i.i.d.) sampling scenario in method design (or evaluation) (Li et al., 2023; Tang et al., 2023; Maini et al., 2021). This means that when the target model employs special but unknown sampling, the sampling mismatch presents a significant challenge for all dataset usage inference techniques.

As noted in Footnote 4, the unbiasedness of our methods hinges on the correlation term $\frac{\text{Corr}_i(\text{TPR}_i - \text{FPR}_i, \gamma_i)}{\text{TPR} - \text{FPR}}$. (Under uniformly random sampling, this term is zero since $\gamma_i = p$ for all $i = 1, 2, \ldots, |X|$.) For settings with special data sampling, we construct subgroups where $\text{TPR}_i - \text{FPR}_i \approx \text{TPR}' - \text{FPR}'$ and debias within each to obtain an unbiased estimator of $\gamma_i$ (see Figure 6 for empirical validation).

To evaluate the performance of all methods under the special sampling challenge, we conduct experiments where the model trainer applies dataset selection techniques (Guo et al., 2022; Sorscher et al., 2022) to select $p$-percentage of the target dataset to include in training. Specifically, we adopt an error-based coreset selection method, EL2N (Paul et al., 2021), to rank the target dataset by "data significance"[6] using 10 trained models that have not seen the target dataset. Then, we constructed the probability vector $\boldsymbol{\alpha}^{(p)}$ over the target dataset $X$ according to this "data significance" for each $p \in \{0.00, 0.05, \ldots, 1.00\}$. The data in the target dataset that used in the training of target models are therefore sampled according to each $\boldsymbol{\alpha}^{(p)}$. By contrast, the reference models are trained under uniformly random $p\%$ subset of the target dataset[7]. These scenarios represent the least favor samplings for our methods, as the EL2N is closely related to $TPR_i - FPR_i$, which introduces strong correlation between $\boldsymbol{\alpha}$ and the errors.

Table 2 shows that all other methods degrade in performance under special sampling challenges and exhibit similar performance across the two symmetric sampling methods. This confirms our earlier discussion on the distribution mismatch challenge from special sampling: they make mistakes occur early or late in the process. Nevertheless, the effectiveness of our debiasing method largely remains and yields better performance than all other methods. By contrast, the MLE method suffers the most significantly under the mismatch, since its superiority comes from fully recover the possible model distribution when data sampling scheme is known.

**Dataset usage estimation error decreases as the ratio $|X|/|D_{\text{Train}}|$ increases.** Previously, we focused on a challenging scenario where the owner's dataset (size 500) constitutes only a small portion of the target model's training set $D_{\text{Train}}$, i.e., the ratio $|X|/|D_{\text{Train}}|$ was small. We now examine the impact of the (relative) dataset size on the DUCI performance. To do this, we evaluated our method using target dataset sizes from 500 to 25,000 (i.e., relative ratios from 0.02 to 1.00), while keeping the full training set fixed at 25,000 examples (half of CIFAR-10's training set).

Table 3 demonstrates that all methods benefit from the increasing relative dataset size. This is because, as the ratio $|X|/|D_{\text{Train}}|$ increases, the randomness from the remaining data in $D_{\text{Train}}$ decreases, making dataset usage inference easier. For instance, in the extreme case where $|X|/|D_{\text{Train}}| = 1$, there is no randomness at 100% utilization. This observation highlights that *prior evaluations in the literature (Li et al., 2023; Maini et al., 2021), which test whether a model was trained on private dataset A versus an alternative dataset B, are oversimplified and offer limited insights into real-world scenarios*. Additionally, we noticed that while the improvement for MIA baselines is modest (as

---

[6]A term we use cautiously, as EL2N can only be considered as a reflection of data value.

[7]Since the sampling method used in the target model is inaccessible, assuming random sampling is the a reasonable (if not the best) strategy a dataset owner can take for simulating the model trainer.

MIA does not leverage dataset-level statistics and benefits only from reduced randomness at larger ratios), our debiasing process sees a more substantial gain. This is because the sampling error in our debiasing is proportional to $\frac{1}{\sqrt{N \cdot |X|}}$ (see formal proof in Appendix G), meaning that as the target dataset size increases, our debiasing procedure becomes more accurate.

**On (the possibility of) leveraging correlation in membership prediction process** Our method implicitly assumes that the prediction of membership probability for one record $i$ does not affect that of another record $j$, i.e., $\mathbb{E}[\hat{p}_i \hat{p}_j] = \mathbb{E}[\hat{p}_i]\mathbb{E}[\hat{p}_j]$. *Will ignoring "correlations" between $\hat{p}_i$ and $\hat{p}_j$ make our method sub-optimal?* We derive a pairwise debiasing approach in Appendix I.2, but observe only minimal gains. To understand this, we measure the covariances between the predictions for pairs of records in the target dataset (no duplication) using the error metric in Equation (34), and find that most pairs show near-zero covariance (Figure 4). This aligns with the prior theoretical findings of small correlation between records (Pillutla et al., 2023). This suggests *limited room* to improve our method by modeling correlations on *natural* dataset.

**Case study: quantifying book copyright infringement** can be seen as a specific case of data usage inference—given a model likely infringing on book copyrights, the authors want to determine the extent of content usage. We conduct experiments on GPT-2 (Radford et al., 2019) and 50 new books with first editions in 2023, unseen by GPT-2, from the BookMIA dataset (Shi et al., 2023). We fine-tune pre-trained GPT-2 on varying proportions of these books to create target models. The sequences in books are sampled **contiguously**, **rather than randomly and uniformly**, from the copyrighted books in order to simulate more realistic scenarios. Results in Table 4 show that both MIA Guess and MIA Score baselines struggle with varying extents of infringement, particularly when infringement is less significant. In contrast, our method estimates the extent of infringement much more reliably, with a maximum MAE (Equation (3)) of 0.168, compared to 0.335 for MIA Guess and 0.5 for MIA Score. However, we observed that, compared to image data, the error variance in text data is generally larger. We defer the discussion and additional experimental details to Appendix J.

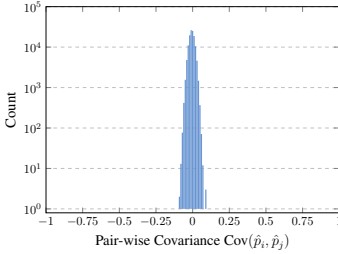

Figure 4: Histogram of the covariance $\mathrm{Cov}(\hat{p}_i, \hat{p}_j)$ between data usage predictions of all data pairs $i$ and $j$ in the target dataset.

Table 4: *(Book copyright infringement)* Mean Absolute Error (MAE) $\mathbb{E}[|\hat{p}_i - p|]$ for all methods under different proportion $p$.

| $p$ | MIA Guess | MIA Score | Our Method |
|---|---|---|---|
| 0.0 | 0.3350 | 0.5019 | **0.1684** |
| 0.2 | 0.2396 | 0.4029 | **0.1573** |
| 0.4 | 0.1202 | 0.2381 | **0.0833** |
| 0.6 | 0.0439 | 0.1028 | **0.0316** |
| 0.8 | 0.1286 | 0.0720 | **0.0453** |
| 1.0 | 0.2448 | 0.2189 | **0.0568** |
| $\max_p$ MAE | 0.3350 | 0.5019 | **0.1684** |

## 6 Conclusions

We identify and formally define the problem of Dataset Usage Cardinality Inference (DUCI), which quantifies dataset utilization—a significant issue that has been overlooked or misinterpreted in prior literature. To solve this, we propose a low-cost, reliable, and unbiased estimator for DUCI, along with confidence intervals for uncertainty. Our extensive experiments demonstrate the effectiveness of our approach, including its practical application in detecting book copyright infringement.

**Limitations and future works** We have thoroughly discussed the special sampling challenges in DUCI in Table 2 and demonstrated improved robustness of our methods compared to baselines. However, there is still room for further enhancement, particularly in achieving statistical guarantees under adversarial samplings. Another promising avenue is applying our methods to related tasks, such as debiasing and aggregating user-level inference (Li et al., 2022a; Chen et al., 2023; Kandpal et al., 2023) for finer-grained neighborhood inference. Given that whether or not datasets similar to copyrighted data, but not directly used in training the target models, should be considered members is still an open question, we leave these extensions for future work. Lastly, while our method is independent of specific MIA and dataset choices, and quantifies uncertainty in proportion prediction, dataset crafting methods that enable FPR guarantees for MIAs could be a vital direction for stronger MIAs and further reducing our method's uncertainty.

ACKNOWLEDGMENT
The authors would like to thank Martin Strobel for his valuable feedback and discussions that contributed to this work, as well as the anonymous reviewers for their insightful comments. Jiayuan Ye is supported by the Apple Scholars in AI/ML PhD fellowship. This research is supported by research awards from Google, and the Ministry of Education, Singapore, under its Academic Research Fund Tier 2 (MOE-T2EP20223-0015).

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

# Appendices Contents

| Notation | Description |
|----------|-------------|
| $T$ | Training algorithm, may have access to auxiliary public data. |
| $\rho$ | Population data pool. |
| $D$ | Training dataset. |
| $X$ | Target dataset (from data owner). |
| $x_i$ | A data record $i$. |
| $m_i$ | Membership indicator (1 if $x_i$ is in training set $D$, otherwise 0). |
| $\hat{m}_i$ | Membership guess from an inference algorithm. |
| $\mathcal{A}$ | Membership inference algorithm. |
| $f_\theta$ | Target model with parameter $\theta$. |
| $f_\theta(x)$ | Output of the target model on $x$. |
| $p$ | Proportion of dataset $X$ usage in training $\theta$. |
| $\hat{p}$ | Estimated dataset usage proportion. |
| $\gamma_i$ | Probability of sample $x_i \in X$ being used in training $f_\theta$. |
| $\hat{p}_i$ | Estimated probability of sample $x_i \in X$ being used in training $f_\theta$. |
| $\theta_j$ | Parameters of reference models ($j = 1, \ldots, N$) used for estimation. |

Table 5: Summary of notations used in the paper.

## A    RELATED WORKS

**Membership inference attack**    Membership inference attacks (MIAs) Ye et al. (2022); Shokri et al. (2017) aim to determine whether a specific data entry is part of the training dataset of a target model. Initially gained traction in medical research Homer et al. (2008), MIAs were first introduced into machine learning by Shokri et al. (2017), subsequently inspiring numerous shadow model attacks Long et al. (2020); Sablayrolles et al. (2019); Song & Mittal (2021); Carlini et al. (2022). These shadow model attacks train a collection of *shadow models* (or *reference models*) to simulate the target model's behavior. All shadow models are trained with the same architecture as the target model using random subsets of data that follow the same distribution as the target model's training data, including or excluding the target data point, to estimate a statistical distribution of certain score function (such as but not limited to loss Sablayrolles et al. (2019); Yeom et al. (2018), confidence Salem et al. (2018), entropy Song & Mittal (2021), or loss trajectory Liu et al. (2022)) reflecting the data point's presence or absence in the training set. Aside from shadow model attacks, an alternative line of research eliminates the need to train multiple shadow models. Instead, they queries the target model with *population data* that is related to, or follows the same distribution as, the target point Wen et al. (2022); Bertran et al. (2023); Zarifzadeh et al. (2024); Jayaraman et al. (2020); Li & Zhang (2021); Long et al. (2018); Xie et al. (2024). These data are utilized as a reference for inferring membership. Individual MIAs cannot be directly adapted to solve the dataset utilization cardinality inference problem because they often cannot achieve perfect distinguishability, and their predictions vary drastically depending on the specific threshold or test used. Therefore, it is necessary to consider the uncertainty in the attack decision to reliably infer the extent of dataset usage. Moreover, the cost of conducting point-wise MIA tests for the entire dataset is another concern, which calls for an efficient adaptation.

### A.0.1    DATASET OWNERSHIP VERIFICATION AND DATASET PROVENANCE TRACING

Dataset Ownership Verification (DOV) and Dataset Provenance Tracing are two related research area to our Dataset Usage Inference problem. Currently, all known dataset ownership verification techniques utilize dataset watermarking, which involves altering a subset of the data to embed a detectable signature Li et al. (2023; 2020); Tang et al. (2023); Li et al. (2022b); Sun et al. (2022); Sablayrolles et al. (2020); Hu et al. (2022); Zou et al. (2022); Wenger et al. (2022); Maini et al. (2021); Song & Shmatikov (2019). In addition to these, a handful of non-watermark-based dataset tracing techniques, although not specifically designed for the dataset ownership problem, can be adapted to address the dataset ownership problem or are related to this issue. These methods primarily

encompass variants of membership inference techniques that operate beyond the individual level Song & Shmatikov (2019); Maini et al. (2021); Li et al. (2022a); Miao et al. (2021). Thi

**Dataset watermarking**   Dataset watermarking Li et al. (2023); Tang et al. (2023); Li et al. (2020); Sun et al. (2022), predominantly utilizing poison-only backdoor attacks (that manipulate only the training data while keeping training components, model parameters, and structures intact Li et al. (2022c)), currently leads in the field of dataset ownership inference. Generally, these methods capitalize on the memorization capabilities of machine learning models during training to embed spurious features (information divergent from the true data distribution) into the model. This embedded knowledge is then used for subsequent verification: the detection of specific backdoor behaviors (targeted or untargeted) in the model's output during the verification phase indicates the model was trained with the protected dataset.

While the earlier watermarking techniques modified the target label of poisoned samples Li et al. (2020); Hu et al. (2022), contemporary and more sophisticated methods employ untargeted backdoor attacks Li et al. (2022b); Sun et al. (2022) or clean-label backdoor attacks Sablayrolles et al. (2020); Tang et al. (2023); Zou et al. (2022), which are significantly more stealthy due to their preservation of label consistency. Several works have extended their focus to different application scenarios, such as protecting open-source code Sun et al. (2022) or performing classification tasks without knowledge of labels Wenger et al. (2022). Additionally, recent works have attempted to minimize the impact on model utility when modifying samples Guo et al. (2023). However, all these modification-based methods are not suitable for scenarios where the goal is to protect the ownership of already released datasets, nor do they guarantee data utility in sensitive scenarios, such as with medical data.

**Dataset Inference**   While several class-level membership inferences Li et al. (2022a); Miao et al. (2021) exist for object detection, such as voice or face recognition from an individual, these works are limited to their specific use cases and are not applicable to the broader scenarios considered in this paper: they require that the data group has a unique pseudo-label. For text data that lacks a clear boundary between records, Song & Shmatikov (2019) utilizes the long tail of the text data distribution within a user's document to carry out user-level MIA. The work most closely related to ours is Dataset Inference Maini et al. (2021), which hypothesizes that private data in general will have a larger distance to the decision boundary than public data, since the model always attempts to maximize this distance during training. It trains a linear binary classifier that takes in the *distance-to-boundary* for each data record to predict its membership. The underlying idea of this method is to use membership inference based on the average of a *loss-like signal* over the data group as a metric. However, all these methods still focus on solving the *all-or-none* binary question, and thus also fail to address the problem of inferring the dataset usage extent.

Another area of research in copyright infringement, such as Vyas et al. (2023), focuses on protecting generative models from producing copyrighted content, regardless of whether the model was trained on copyrighted data. Although this task may appear similar to ours, it focuses on different aspect of copyright and can be addressed quite differently—for example, by adding distance constraints to prevent outputs from closely resembling protected content. In contrast, our goal is to infer properties of a target model's training set based on its outputs. As this line of work is greatly distinct from ours, we did not discuss them in the scope of this paper.

## B IMPLEMENTATION DETAILS FOR WATERMARKING AND DATASET INFERENCE

For both implementations, the target protected dataset $X$ (for which we infer the usage) is of size 500. All models are Wide Residual Networks trained on 25,000 records from CIFAR-10 (that contains a randomly sampled fraction of the target dataset). This setting ensures that the target protected dataset is only a small subset of the target model's training set, making it a more practical scenario.

### B.1 WATERMARKING

The watermarking baseline depicted in Figure 1 is an implementation of Algorithm 1 in (Li et al., 2023). Our objective is merely to demonstrate the limitations of watermarking methods in addressing the Dataset Usage Cardinality Inference (DUCI) problem. Consequently, we selected the most robust and straightforward backdoor watermarking techniques, disregarding stealthiness. We employed the targeted poisoned-label attack with a simple white square as the trigger pattern (examples are shown in Figure 5). Considering the sub-sampling inherent in the DUCI problem, we opted for a poison rate of 30% for the protected dataset $X$ with a size of 500 (lower poison rates were tested, but the likelihood of a successful backdoor attack diminishes with reduced poison rates). We designated class 0 as the target label.

The following steps were undertaken:

1. We randomly sampled $m = 2000$ instances from the CIFAR10 test set, adhering to the methodology outlined in the original paper. Specifically, we only chose instances correctly classified by the benign model to minimize the impact of the model's intrinsic accuracy on the test. If the model exhibits low accuracy, it might erroneously categorize benign instances as poisoned, thereby influencing the confidence value.

2. Following Li et al. (2023), we obtained the confidence score at the ground-truth label for each benign sample $x$ and its poisoned counterpart $x'$ from the target model, denoted as $P_b = f(x)$ and $P_p = f(x')$, respectively.

3. We conducted pairwise T-tests: For each pair of samples, the difference in posterior probabilities $P_b - P_p$ was calculated. Utilizing these differences, a pairwise left tail T-test was performed to test the null hypothesis $H_0 : P_b = P_p + \tau$, where $\tau$ is a hyperparameter. We evaluated performance across different values of $\tau$ for $\tau \in \{0.0, 0.005, \ldots, 0.02\}$. The results are reported under the best-performing $\tau$.

### B.2 DATASET INFERENCE

We implemented a simplified yet effective version of the approach described by Maini et al. (2021), replacing the random walk distance (to the decision boundary) with logits queried on the target model for each data point. Empirically, we observed no significant performance degradation with this simplification, and the essential principle remains intact[8]. This approach is based on a hypothesis that private data in general will have a larger distance (smaller loss and higher prediction confidence) to the decision boundary than public data, since the model always attempts to maximize this distance during training. Therefore, comparing the average distance/logits score across the entire data group with that of public data can help to make a decision. All subsequent steps adhere strictly to those outlined in the original paper:

1. Randomly sample $m$ data points each time from both the public dataset $X$ and private dataset, and query the target model to compute their confidence scores, $c$ (public) and $c_v$ (private).

2. We repeated Step 1 for $|X|/m$ iterations, calculating the mean values $\mu = \bar{c}$ and $\mu_v = \bar{c_v}$.

3. We conducted the pairwise T-test on the distributions of $c$ and $c_v$, calculating the p-value to test the null hypothesis (the dataset is not being used) $H_0 : \mu = \mu_v$.

---

[8]For example, the distance-to-boundary in a linear model is equivalent to the loss

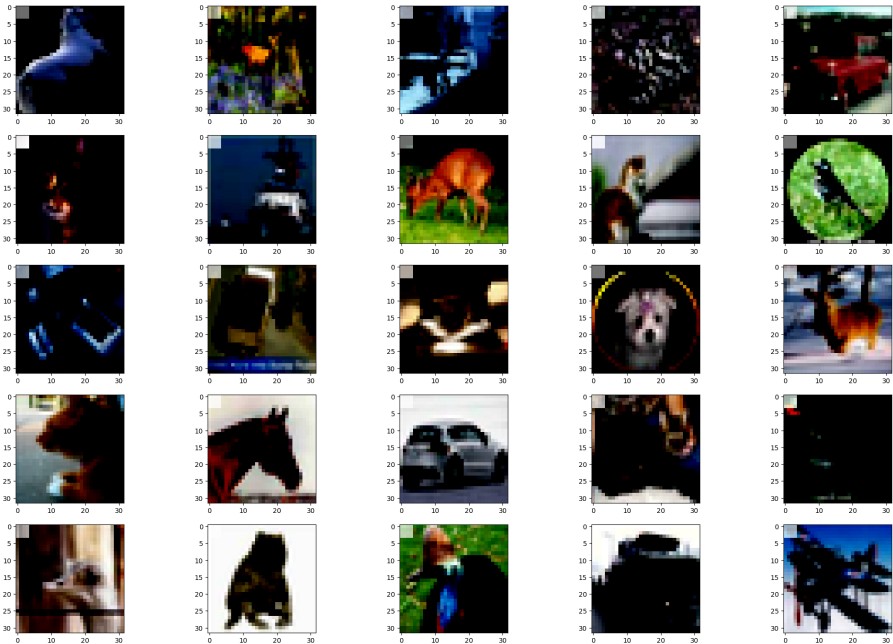

Figure 5: Visualization of poisoned data. Following standard method, we add the white square with random transparency as the trigger pattern.

## C IMPLEMENTATION DETAILS OF OUR METHOD

We next introduce the implementation details of our method.

Algorithm 1 provides the pipeline of our method, where we use the existing membership inference algorithm, denoted as MIA, as the membership identification method to provide membership guesses $\hat{m}_i$ for all $x_i \in X$. We first trained $N$ reference models, $\theta_1, \ldots, \theta_N$, such that each reference models are trained on half of the dataset $X$. Knowing the membership status of all $x_i \in X$ for these reference models, we can run MIA on them to get membership guesses for each $x_i$, and then compute the TPR and FPR as defined in Equation (6) and Equation (7). The aggregation of debiased membership guesses gives us an unbiased estimator of the expected inclusion proportion. *To reduce potential errors from correlations between the MIA score and membership predictions in certain MIA methods, we recommend using separate reference models as the target model and attack models during the debiasing process.*

### C.1 SELECTION OF MEMBERSHIP INFERENCE ALGORITHMS

While any membership identification method capable of providing membership guesses can, in principle, be applied to our approach, we use Membership Inference Attacks (MIA) in our experiments. Specifically, we consider two state-of-the-art MIA algorithms (and their variants): the Likelihood Ratio Attack (LiRA) (Carlini et al., 2022) and the Robust Membership Inference Attack (RMIA) (Zarifzadeh et al., 2024)[9] Given that both methods perform similarly with practical computational budget, and RMIA shows greater robustness under extremely limited computational resources, we present the RMIA results in the main paper and include results for both methods in the appendix.

---

[9]RMIA has two simplified versions with lower computational costs: the population attack (which omits the normalization denominator) and the reference-model attack (which does not use population data). For the text data experiment in Appendix J, we use the reference-model attack due to the high inference cost of querying population data on language models, as done in Meeus et al. (2025). For DUCI with a single reference model, we use the population attack to avoid strong membership-score correlations introduced by reference models.

---

**Algorithm 1 Dataset Usage Inference:** Infer the proportion $p$ of data in a dataset $X$ being used in the training of a given target model $\theta$

---

**Require:** Dataset $X$, Training algorithm $\mathcal{T}$ (including the data distribution $\mathbb{D}$), Membership inference algorithm MIA, Target model $\theta$ (black-box access)

    *Train reference models for debiasing (as few as one model)*

1: **for** $j = 1$ to $N$ **do**
2:     $D \leftarrow \mathbb{D}$
3:     $\boldsymbol{m}^{(j)} \leftarrow \text{Bernoulli}(\frac{1}{2})^{|X|}$
4:     $\theta_j \leftarrow \mathcal{T}(D \cup \{\boldsymbol{m}^{(j)} \odot X\})$ *Select data from $X$ by $\boldsymbol{m}^{(j)}$*
5: **end for**
6: Save $\boldsymbol{m}^{1,\ldots,N} = \{\boldsymbol{m}_1, \ldots, \boldsymbol{m}_N\}$

    *Debias*

7: Run MIA for $x_i \in X$ on $\theta_j$ for $j \in 1 \ldots, \theta_N$ to estimate $\Pr(\hat{m} = 1 | m = 0)$ and $\Pr(\hat{m} = 1 | m = 1)$:

$$\Pr(\hat{m} = 1 | m = 0) = 2 \sum_{j=1}^{N} \sum_{i=1}^{|X|} \frac{\mathbf{1}_{\text{MIA}(\theta_j, x_i) = 1 \text{ and } \boldsymbol{m}_i^{(j)} = 0}}{N \cdot |X|}$$

$$\Pr(\hat{m} = 1 | m = 1) = 2 \sum_{j=1}^{N} \sum_{i=1}^{|X|} \frac{\mathbf{1}_{\text{MIA}(\theta_j, x_i) = 1 \text{ and } \boldsymbol{m}_i^{(j)} = 1}}{N \cdot |X|}$$

    *Launch datase usage inference*

8: **for** $i = 1$ to $|X|$ **do**
9:     $\hat{m}_i = \text{MIA}(x_i; \theta)$ *Get individual membership guess on the target model*
10:     Debiasing each individual membership guess $\hat{m}_i$:

$$\hat{p}_i = \frac{\hat{m}_i - \Pr(\hat{m} = 1 | m = 0)}{\Pr(\hat{m} = 1 | m = 1) - \Pr(\hat{m} = 1 | m = 0)}$$

11: **end for**
12: Output the proportion estimation $\hat{p} = \sum_{i=1}^{|X|} \hat{p}_i$

---

**Likelihood Ratio Attack (LiRA)** Define the scaled logits for a point $x$ of a model $\theta$ as $\text{logit}(x;\theta) = \log\left(\frac{f_\theta(x)}{1-f_\theta(x)}\right)$, where $f_\theta(x)$ is the output probability at the ground-truth class. LiRA (Carlini et al., 2022) generates a membership guess $\hat{m}_i$ for a point $x_i$ by conducting a likelihood ratio test, utilizing scaled logits as the test signal, as follows:

$$\hat{m}_i = \text{MIA}(x_i;\theta) = \mathbb{1}\left[\frac{\Pr\left(\text{logit}(x_i;\theta)|\mathcal{N}(\mu_{in},\sigma_{in}^2)\right)}{\Pr\left(\text{logit}(x_i;\theta)|\mathcal{N}(\mu_{out},\sigma_{out}^2)\right)} > \beta(x_i)\right] \tag{11}$$

where $\mathcal{N}(\mu_{in},\sigma_{in}^2)$ and $\mathcal{N}(\mu_{out},\sigma_{out}^2)$ are the Gaussian distributions of $\text{logit}(x_i;\theta)$ estimated on the $N$ reference models trained with (in) and without (out) $x_i$, respectively.

The essential idea of LiRA is to compare the likelihood of observing the target model's signal given the distribution of models trained with the target point $x$ against the distribution of models trained without $x$, where both distributions are estimated as Gaussian.

**Robust Membership Inference Attack (RMIA)**

$$\hat{m}_i = \text{MIA}(x_i;\theta) = \mathbb{1}\left[P_{z\sim\rho}(LR_\theta(x_i,z) \geq 1) \geq \beta(x_i)\right]$$

$$LR_\theta(x_i,z) = \left(\frac{\Pr(x_i|\theta)}{\Pr(x_i)}\right)\cdot\left(\frac{\Pr(z|\theta)}{\Pr(z)}\right)^{-1} \tag{12}$$

RMIA (Zarifzadeh et al., 2024) compares the target point $x_i$ with a number of population data $z \sim \rho$ from the same distribution, using the ratio $\frac{\Pr(x_i|\theta)}{\Pr(x_i)}$ as signal, where $\Pr(x_i|\theta)$ is the prediction score (SoftMax) of output of the target model and $\Pr(x_i)$ is the normalization term which is calculated by averaging $\Pr(x_i|\theta_j)$ over all reference models $\theta_j$ for $j = 1,2,\ldots,2N$ trained with or without $x_i$, i.e., $\Pr(x_i) = \frac{1}{N}\sum_{i=1}^{N}\Pr(x_i|\theta_j) + \frac{1}{N}\sum_{i=N}^{2N}\Pr(x_i|\theta_j)$ where $\theta_1,\theta_2,\ldots,\theta_{N-1}$ are models trained with $x_i$ and $\theta_N,\theta_{N+1},\ldots,\theta_{2N}$ are models not trained with $x_i$. For the offline setting, the $\Pr(x_i)$ can be estimated only using models not trained with $x_i$, i.e., $\theta_N,\theta_{N+1},\ldots,\theta_{2N}$. It can be extended to launch membership inference using single reference model.

## C.2 LiRA IMPLEMENTATION

Given the target points $x \in X$ and target model $\theta$, we implement the LiRA following Carlini et al. (2022) as follows:

1. Train $2N$ reference models. Each are trained on uniformly and randomly sampled half of $X$ together with uniformly and randomly sampled half of the population pool. Therefore, each $x_i \in X$ will have $N$ reference models trained with it and the remaining $N$ reference models trained without it.

2. For each $x_i$, estimate two Gaussian distributions on reference models: one distribution on models trained with data record $x_i$ (denoted as $\mathcal{N}(\mu_{in},\sigma_{in}^2)$) and another distribution on models trained without $x_i$ (denoted as $\mathcal{N}(\mu_{out},\sigma_{out}^2)$). The mean $\mu$ and variance $\sigma^2$ of each distributions are estimated on the scaled logits of each data record $x_i \in X$ over $N$ reference models (trained with $x$ or without $x$, respectively).

3. For each data record $x_i$, querying the target model to get the scaled logit $\text{logit}(x_i;\theta)$ and then compute the likelihood of observing it on two distributions: $\Pr\left(\text{logit}(x_i;\theta)|\mathcal{N}(\mu_{in},\sigma_{in}^2)\right)$ and $\Pr\left(\text{logit}(x_i;\theta)|\mathcal{N}(\mu_{out},\sigma_{out}^2)\right)$.

4. Compute the likelihood ratio $\text{LR}(x_i) = \frac{\Pr\left(\text{logit}(x_i;\theta)|\mathcal{N}(\mu_{in},\sigma_{in}^2)\right)}{\Pr\left(\text{logit}(x_i;\theta)|\mathcal{N}(\mu_{out},\sigma_{out}^2)\right)}$ for each data record $x_i$.

5. $\hat{m}_i = \mathbb{1}(\text{LR}(x_i) > \beta)$.

## C.3 RMIA IMPLEMENTATION

In the original RMIA paper (Zarifzadeh et al., 2024), the population data utilized in Equation (12) could be other target data. Integrating RMIA scores directly into our aggregation approach would encounter issues of the differential rate of change as the true proportion of utilized data rises. To

address this, we implemented the following adaptive version of RMIA focusing on a fixed target dataset: Similarly, given the target models $\theta$ and the target points $x_i \in X$, we implement the RMIA as follows:

1. Train $N$ reference models. Each are trained on uniformly and randomly sampled half of $X$ together with uniformly and randomly sampled half of the population pool. Therefore, each $x_i \in X$ will have $N_1$ reference models trained with it and the remaining $N_2 = N - N_1$ reference models trained without it. (When $N \geq 2$, we can ensure $N_1 = N_2$. When $N = 1$, we can do approximation as described below.)

2. For each $x_i \in X$, denote the reference models $\theta_1, \theta_2, \ldots, \theta_{N_1}$ as the models trained with $x_i$, and $\theta_{N_1+1}, \theta_{N_1+2}, \ldots, \theta_{N_1+N_2}$ as the models trained without $x_i$.

   (a) ($N >= 2$) Compute $\frac{\Pr(x_i|\theta)}{\Pr(x_i)} = \frac{f_\theta(x_i)}{\frac{1}{N_1}\sum_{j=1}^{N_1} f_{\theta_j}(x_i) + \frac{1}{N_2}\sum_{j=N_1+1}^{N} f_{\theta_j}(x_i)}$, where $f_\theta(x_i)$ are the model's confidence value at the ground-truth class.

   (b) ($N = 1$) We can use an linear approximation to compute the denominator. Assume $x_i$ is not in the training set of the only reference model $\theta_j$, we can estimate the probability of a model trained with it as $af_{\theta_j}(x_i) + 1 - a$. A similar approach applies in the reverse case. We select $a = 0.3$ according to Figure 6 in (Zarifzadeh et al., 2024). In this case, only $N$ reference models are required for inference. (Note that the same $N$ models can be shared among all $x_i$ as the models trained without $x_i$.)

3. (Optional) For $t = 1, 2, \ldots, T$:

   (a) Select a $z^t$ from the population pool such that it is in the training set of $N$ reference models and not in the training set of the remaining $N$ reference models. Note that $z^t \notin X$ since it is selected from population pool.

   (b) Compute $\frac{\Pr(z^t|\theta)}{\Pr(z^t)} = \frac{f_\theta(z^t)_{y_z}}{\frac{1}{N}\sum_{i=1}^{N} f_{\theta_{z^t}}(z^t)_{y_z} + \frac{1}{N}\sum_{i=1}^{N} f_{\theta_{-z^t}}(z^t)_{y_z}}$.

   (c) Compute $\mathrm{LR}(x_i, z^t) = \left(\frac{\Pr(x_i|\theta)}{\Pr(x)}\right) \cdot \left(\frac{\Pr(z^t|\theta)}{\Pr(z^t)}\right)^{-1}$ for $(x_i, z^t)$ pair.

4. For each $x_i \in X$, compute $P_z(\mathrm{LR}(x_i, z) > 1) = \frac{\mathbb{1}[\mathrm{LR}(x_i, z^t) > 1]}{T}$

5. $\hat{m}_i = \mathbb{1}(P_z(\mathrm{LR}(x_i, z) > 1) > \beta)$.

## C.4 SELECTION OF THRESHOLDS $\beta$

For the choice of threshold in both LiRA and RMIA, we sweep over all possible threshold to select the $\beta$ as the optimal threshold maximizing the Youden's index ($J = \mathrm{TPR} - \mathrm{FPR}$) Youden (1950) on reference modes, which provides the best balance between sensitivity (TPR) and specificity (1 - FPR). Depends on the computation of TPR and FPR in dataset usage cardinality inference problem, the threshold $\beta$ can be computed either in the *individual-level* or *dataset-level*.

*Individual-level threshold:*

$$\hat{\beta}_i = \arg\max_\beta [\Pr(\hat{m}_\beta(x_i; \theta) = 1|m = 1) - \Pr(\hat{m}_\beta(x_i; \theta) = 1|m = 0)] \tag{13}$$

Here, $\Pr(\hat{m}_\beta(x_i; \theta) = 1|m = 1)$ is the True Positive Rate (TPR) computed under threshold $\beta$ for each data point $x_i$, and $\Pr(\hat{m}_\beta(x_i; \theta) = 1|m = 0)$ is the False Positive Rate (FPR), similarly computed. Each of them is estimated on reference models.

*Dataset-level threshold:* The individual threshold may not be optimal when the number of samples is limited. In such cases, we can adopt dataset-level thresholds:

$$\hat{\beta} = \arg\max_\beta [\Pr_X(\hat{m}_\beta = 1|m = 1) - \Pr_X(\hat{m}_\beta = 1|m = 0)] \tag{14}$$

where $\Pr_X(\hat{m}_\beta = 1|m = 1)$ is the True Positive Rate (TPR) as depicted in Equation (7) under threshold $\beta$, and $\Pr(\hat{m}_\beta(x; \theta) = 1|m = 0)$ is the False Positive Rate (FPR) in Equation (6), computed in the same setting.

Our all results presented in the main paper are based on dataset-level threshold and dataset-level debiasing due to the smaller sampling error under limited reference models.

### C.5 EFFECT OF MIA CHOICES

While any membership identification method that provides membership guesses can be integrated into our approach, we empirically found that **methods with more stable performance against training randomness (e.g., less sensitivity to initialization and data variability) tend to perform better under our debiasing framework.**

For example, RMIA combined with debiasing outperforms LiRA combined with debiasing on image classification models, particularly when the number of reference models is small (Figure 2). This may be because LiRA relies solely on reference models for membership inference, making it more susceptible to variability across trials. In contrast, RMIA explicitly incorporates population data in its test, in addition to reference models, leading to greater stability and lower prediction errors—especially when the number of reference models is limited.

However, for text generation models, RMIA without population data (i.e., the naive reference model attack) performs better than RMIA with population data probably due to the normalization effect. This suggests that *a more powerful attack does not necessarily yield better performance under debiasing. Instead, weaker yet more stable attacks (i.e., those producing comparable scores across models) benefit more from DUCI.*

## D IMPOSSIBILITY OF MULTIPLICATIVE GUARANTEE FOR DUCI

*Why use additive error instead of multiplicative error?* Based on the standard packing argument (Hardt & Talwar, 2010), providing a multiplicative guarantee for DUCI is fundamentally impossible, regardless of the method used. A simple example illustrates this: in the extreme case where the ground truth $p = 0$, the estimate $\hat{p}$ must also be 0 to keep the multiplicative ratio bounded. Below, we provide a more detailed explanation.

Let the ground-truth membership probability for record $i$ in the training dataset be $\gamma_i$. By definition, a dataset sampled under probabilities $(\gamma_1, \cdots, \gamma_n)$ and $(\gamma_1 + \frac{1}{10n}, \ldots, \gamma_n + \frac{1}{10n})$ can be identical with at least constant probability $\frac{9}{10}$ by the union bound. Consequently, with constant probability, a DUCI algorithm cannot reliably distinguish between datasets sampled with $(\gamma_1, \ldots, \gamma_n)$ and $(\gamma_1 \pm \frac{1}{n}, \ldots, \gamma_n \pm \frac{1}{n})$, leading to an unavoidable additive error of $\frac{1}{n}$. Thus, for any fixed $c \geq 1$, as $\gamma_i \to 0$, either the multiplicative error $\frac{\hat{p}_i}{\gamma_i}$ diverges to infinity or the multiplicative error $\frac{\hat{p}_i}{\gamma_i + \frac{1}{n}}$ shrinks to zero, falling outside the range $(1/c, c)$.

On the other hand, as discussed in Figure 3, additive error provides a more consistent and meaningful metric for DUCI. Since DUCI is inherently a discrete counting problem, the focus is on incorrect counts, making additive error a more appropriate measure. For instance, in a protected dataset of size 10, an additive error of 0.1 consistently represents a single misprediction. In contrast, multiplicative error leads to misleading results: a single misprediction when $p = 0.1$ produces the same ratio as mispredicting ***all*** points when $p = 1.0$, which is clearly unreasonable.

# E   ANALYSIS OF POTENTIAL ERRORS FROM SIMPLIFYING RECORD-LEVEL DEBIASING TO DATASET-LEVEL DEBIASING

In Equation (4), we demonstrated that applying a record-level debiasing process to membership predictions before aggregation yields an unbiased estimator, $\hat{p} = \frac{1}{|X|} \sum_{i=1}^{|X|} \hat{p}_i$ for the ground-truth proportion $p$. However, due to the computational cost and notable sampling error when the number of reference models is small, we instead estimate the TPR and FPR across the entire dataset $X$. We next show that the simplification in the debiasing process will not introduce errors in many practical sampling scenarios, such as uniform or i.i.d. sampling (which are the common scenario considered in the long line of prior works in binary dataset inference literature listed in the Introduction section). Only in special cases where there is a strong correlation between the probability of sampling $i$-th point $\gamma_i$ and its $\text{TPR}_i - \text{FPR}_i$, there would be a error. However, this can be effectively mitigated by subgroup debiasing, as shown in Table 2.

In Equation (6) and Equation (7), we leverage dataset-level TPR and FPR (i.e., $\text{TPR} = \frac{1}{|X|} \sum_i \text{TPR}_i$ and $\text{FPR} = \frac{1}{|X|} \sum_i \text{FPR}_i$) to replace the individual $\text{TPR}_i$ and $\text{FPR}_i$. This simplification avoids the computationally cost (or large sampling errors) of debiasing each $\hat{p}_i$. This simplification is justified because the proportion $p$ is a dataset-level statistic, as analyzed below.

To avoid confusion, we introduce $\tilde{p}$ and $\tilde{p}_i$ as the estimators under dataset-level debiasing defined by

$$\tilde{p} = \frac{1}{|X|} \sum_i \tilde{p}_i \text{ where } \tilde{p}_i = \frac{\hat{m}_i - \text{FPR}}{\text{TPR} - \text{FPR}}. \tag{15}$$

We next prove $\tilde{p}$ is an unbiased estimator of $p$ whenever a correlation term between $\text{TPR}_i - \text{FPR}_i$ and $\gamma_i$ is zero: by definition (Equation (15)), we compute that

$$\mathbb{E}[\tilde{p}] = \mathbb{E}\left[\frac{1}{|X|} \sum_i \tilde{p}_i\right] = \frac{1}{|X|} \sum_i \mathbb{E}[\tilde{p}_i] = \frac{1}{|X|} \sum_i \mathbb{E}\left[\frac{\hat{m}_i - \text{FPR}}{\text{TPR} - \text{FPR}}\right] \tag{16}$$

Plugging Equation (5) into Equation (16), we can get:

$$\mathbb{E}[\tilde{p}] = \frac{1}{\text{TPR} - \text{FPR}} \cdot \frac{1}{|X|} \sum_i (\gamma_i \cdot \text{TPR}_i + (1 - \gamma_i) \cdot \text{FPR}_i - \text{FPR}) \tag{17}$$

$$= \frac{\frac{1}{|X|} \sum_i [\gamma_i \cdot (\text{TPR}_i - \text{FPR}_i)]}{\frac{1}{|X|} \sum_i [\text{TPR}_i - \text{FPR}_i]} \tag{18}$$

Given $\mathbb{E}[p] = \frac{1}{|X|} \sum_i \gamma_i$, note that

$$\frac{1}{|X|} \sum_i [\gamma_i \cdot (\text{TPR}_i - \text{FPR}_i)] = \left(\frac{1}{|X|} \sum_i \gamma_i\right) \cdot \left(\frac{1}{|X|} \sum_i (\text{TPR}_i - \text{FPR}_i)\right)$$

$$+ \underbrace{\frac{1}{|X|} \sum_i \left(\left(\gamma_i - \frac{1}{|X|} \sum_i \gamma_i\right) \cdot \left(\text{TPR}_i - \text{FPR}_i - \frac{1}{|X|} \sum_i (\text{TPR}_i - \text{FPR}_i)\right)\right)}_{\text{Corr}_i(\text{TPR}_i - \text{FPR}_i, \gamma_i)}$$

(Here, we use the term "correlation" informally to refer to an empirical alignment between $\text{TPR}_i - \text{FPR}_i$ and $\gamma_i$. This is not a statistical correlation between random variables, but rather a covariance-like quantity computed over deterministic values.) By further plugging definition for TPR and FPR in Equation (5) into equation 19 and Equation (16), we compute that:

$$\mathbb{E}[\tilde{p}] = \mathbb{E}[p] + \frac{\text{Corr}_i(\text{TPR}_i - \text{FPR}_i, \gamma_i)}{\text{TPR} - \text{FPR}}. \tag{19}$$

The correlation term suggests that for many practical sampling methods (e.g., sampling without replacement, i.i.d. Poisson sampling), this simplification results in an unbiased estimator for expected utilization proportion because the correlation is 0. For special sampling methods, we can estimate TPR and FPR over subgroups of (rather than the entirety of) dataset $X$ – this ensures (empirical)

unbiasedness, as discussed in Table 2, as long as $\text{TPR}_i - \text{FPR}_i$ is constant within each subgroup. This ensures that the correlation term for each subgroup is 0, providing a group-level debiasing approach. Note that the term "correlation" (slightly abused here) in the context describes how the value of $\gamma_i$'s, a set of prefixed constants in the DUCI pipeline, is determined. This is distinct from the correlation between membership predictions in Figure 4 over random trials of the DUCI pipeline.

# F  DERIVATION OF THE CONFIDENCE INTERVAL

Modern machine learning algorithms and their data sampling processes are highly randomized. To capture the intrinsic uncertainty and confidence in dataset usage inference, we resort to the Lyapunov Central Limit Theorem (CLT) Billingsley (2017) to compute confidence intervals for our estimation method via aggregate statistics. We first introduce the Lyapunov CLT.

**Definition F.1** (Lyapunov condition Billingsley (2017)). *If a sequence of independent random variables* $\{X_i\}_{i=1}^n$ *satisfy* $\mathbb{E}[X_i] = \mu_i < \infty$, $\mathbb{E}[(X_i - \mu_i)^2] = \sigma_i^2 < \infty$, *and for some* $\delta > 0$:

$$\lim_{n \to \infty} \frac{1}{s_n^{2+\delta}} \sum_{i=1}^n \mathbb{E}\left[|X_i - \mu_i|^{2+\delta}\right] = 0, \tag{20}$$

*where* $s_n^2 = \sum_{i=1}^n \sigma_i^2 > 0$, *the Lyapunov condition is said to hold for* $\{X_i, i = 1, \dots, n\}$.

**Theorem F.2** (Lyapunov Central Limit Theorem Billingsley (2017)). *Let* $\{X_i\}_{i=1}^n$ *be a sequence of independent random variables with means* $\mu_i = \mathbb{E}[X_i]$ *and variances* $\sigma_i^2 = Var(X_i)$. *If the Lyapunov condition is satisfied, the following sum converges in distribution to a standard normal distribution:*

$$\frac{\sum_{i=1}^n (X_i - \mu_i)}{s_n} \xrightarrow{d} \mathcal{N}(0, 1).$$

*where* $s_n = \sqrt{\sum_{i=1}^n \sigma_i^2}$ *and* $\xrightarrow{d}$ *denotes the convergence in distribution.*

Theorem F.2 states that if the individual statistics $\hat{p}_1, \cdots, \hat{p}_{|X|}$ in Equation (4) are independent (without needing to be sampled from the same distribution), and the Lyapunov condition is satisfied, their aggregation $\sum_{i=1}^{|X|} \hat{p}_i$ converge to a Gaussian distribution as $|X|$ increases. Therefore, by applying Lyapunov CLT, we obtain the approximate 95% confidence interval for estimating $\mathbb{E}[p]$.

$$\hat{p} - t_{\alpha/2}\sqrt{\frac{s_{|X|}^2}{|X|}} \leq \mathbb{E}[p] \leq \hat{p} + t_{\alpha/2}\sqrt{\frac{s_{|X|}^2}{|X|}}. \tag{21}$$

Here, $\hat{p} = \frac{1}{|X|}\sum_{i=1}^{|X|}\hat{p}_i$, $\alpha = 0.05$, $t_{\alpha/2}$ is the $t$ critical value, and $s_{|X|}^2 = \frac{1}{|X|}\sum_{i=1}^{|X|}\sigma_i^2$, where $\sigma_i^2$ is the population variance of $\hat{p}_i$ which can be empirically estimated over the random trials.

However, considering the cost of launching this confidence interval inference, we porpose to use the sample variance to overestimate $\frac{s_{|X|}^2}{|X|}$ (see proof in Appendix F.1). This gives us the following approximate 95% confidence interval for estimating $p$.

$$\hat{p} - t_{\alpha/2}\sqrt{\frac{s^2}{|X|}} \leq \mathbb{E}[p] \leq \hat{p} + t_{\alpha/2}\sqrt{\frac{s^2}{|X|}}. \tag{22}$$

Similarly, again here $\hat{p} = \frac{1}{|X|}\sum_{i=1}^{|X|}\hat{p}_i$, $\alpha = 0.05$, $t_{\alpha/2}$ is the $t$ critical value, and $s^2 = \frac{1}{|X|-1}\sum_{i=1}^{|X|}(\hat{p}_i - \hat{p})^2$, where $\hat{p} = \frac{1}{|X|}\sum_{i=1}^{|X|}\hat{p}_i$.

## F.1  PROOF OF OVERESTIMATION $\mathbb{E}[s^2] \geq \frac{\sum_{i=1}^{|X|}\sigma_i^2}{|X|}$

**Lemma F.3.** *Given a set of independent random variables* $\hat{p}_1, \cdots, \hat{p}_{|X|}$ *with variances* $\sigma_1^2, \cdots, \sigma_{|X|}^2$ *(which may not be identically distributed), the sample variance* $s^2$ *defined as*

$$s^2 = \frac{1}{|X|-1}\sum_{i=1}^{|X|}(\hat{p}_i - \hat{p})^2 \quad where \quad \hat{p} = \frac{1}{|X|}\sum_{i=1}^{|X|}\hat{p}_i$$

*is at least as large as the average variance* $\frac{\sum_{i=1}^{|X|}\sigma_i^2}{|X|}$.

*Proof.* We start from the expectation of the squared differences:

$$\mathbb{E}[(\hat{p}_i - \hat{p})^2] = \mathbb{E}[\hat{p}_i^2] - 2\mathbb{E}[\hat{p}_i\hat{p}] + \mathbb{E}[\hat{p}^2] \tag{23}$$

We next analyze each term.

For the first term, we have

$$\mathbb{E}[\hat{p}_i^2] = \sigma_i^2 + \mu_i^2 \tag{24}$$

where $\mu_i = \mathbb{E}[\hat{p}_i]$ for any $i$.

For the second term, we expand it as

$$\mathbb{E}[\hat{p}_i \hat{p}] = \mathbb{E}\left[ \hat{p}_i \cdot \frac{1}{|X|} \sum_{j=1}^{|X|} \hat{p}_j \right] = \frac{1}{|X|} \sum_{j=1}^{|X|} \mathbb{E}[\hat{p}_i \hat{p}_j] \tag{25}$$

$$= \frac{1}{|X|} \left( \mathbb{E}[\hat{p}_i^2] + \sum_{j \neq i} \mathbb{E}[\hat{p}_i]\mathbb{E}[\hat{p}_j] \right) \tag{26}$$

$$= \frac{1}{|X|} \left( \sigma_i^2 + \mu_i^2 + \mu_i \sum_{j \neq i} \mu_j \right) \tag{27}$$

$$= \frac{1}{|X|} \left( \sigma_i^2 + \mu_i^2 + \mu_i \left( \sum_{j=1}^{|X|} \mu_j - \mu_i \right) \right) \tag{28}$$

$$= \frac{1}{|X|} \left( \sigma_i^2 + \mu_i \sum_{j=1}^{|X|} \mu_j \right) \tag{29}$$

where equation 26 is by computing $\mathbb{E}[\hat{p}_i \hat{p}_j]$ as follows.

- If $i = j$, $\mathbb{E}[\hat{p}_i \hat{p}_i] = \mathbb{E}[\hat{p}_i^2] = \sigma_i^2 + \mu_i^2$.
- If $i \neq j$, $\mathbb{E}[\hat{p}_i \hat{p}_j] = \mathbb{E}[\hat{p}_i]\mathbb{E}[\hat{p}_j] = \mu_i \mu_j$.

For the third term, by definition, we have

$$\mathbb{E}[\hat{p}^2] = \text{Var}(\hat{p}) + (\mathbb{E}[\hat{p}])^2 = \frac{1}{|X|^2} \sum_{j=1}^{|X|} \sigma_j^2 + \left( \frac{1}{|X|} \sum_{j=1}^{|X|} \mu_j \right)^2 \tag{30}$$

By plugging equation 24, equation 29 and equation 30 into equation 23, we prove that

$$\mathbb{E}[(\hat{p}_i - \hat{p})^2] = \sigma_i^2 + \mu_i^2 - \frac{2}{|X|} \left( \sigma_i^2 + \mu_i \sum_{j=1}^{|X|} \mu_j \right) + \frac{1}{|X|^2} \sum_{j=1}^{|X|} \sigma_j^2 + \left( \frac{1}{|X|} \sum_{j=1}^{|X|} \mu_j \right)^2 \tag{31}$$

Thus

$$\sum_{i=1}^{|X|} \mathbb{E}[(\hat{p}_i - \hat{p})^2] = \sum_{i=1}^{|X|} \sigma_i^2 + \sum_{i=1}^{|X|} \mu_i^2 - \sum_{i=1}^{|X|} \frac{2}{|X|} \left( \sigma_i^2 + \mu_i \sum_{j=1}^{|X|} \mu_j \right)$$

$$+ \sum_{i=1}^{|X|} \left( \frac{1}{|X|^2} \sum_{j=1}^{|X|} \sigma_j^2 + \left( \frac{1}{|X|} \sum_{j=1}^{|X|} \mu_j \right)^2 \right)$$

$$= \left( 1 - \frac{1}{|X|} \right) \sum_{i=1}^{|X|} \sigma_i^2 + \sum_{i=1}^{|X|} \mu_i^2 - \frac{1}{|X|} \left( \sum_{j=1}^{|X|} \mu_j \right)^2 \tag{32}$$

Recognize that by Cauchy–Schwarz inequality, the terms involving $\mu_i$ can be lower bounded as follows.

$$\sum_{i=1}^{|X|} \mu_i^2 - \frac{1}{|X|} \left( \sum_{i=1}^{|X|} \mu_i \right)^2 \geq \sum_{i=1}^{|X|} \mu_i^2 - \frac{1}{|X|} \cdot |X| \cdot \sum_{i=1}^{|X|} \mu_i^2 = 0$$

Thus Equation (32) can be immediately written as

$$\sum_{i=1}^{|X|} \mathbb{E}[(\hat{p}_i - \hat{p})^2] \geq \left( 1 - \frac{1}{|X|} \right) \sum_{i=1}^{|X|} \sigma_i^2$$

Thus by the definition of $s^2$ and by the additivity of expectation, we compute expected value of the sample variance as follows.

$$\mathbb{E}[s^2] = \frac{1}{|X|-1} \sum_{i=1}^{|X|} \mathbb{E}[(\hat{p}_i - \hat{p})^2]$$

$$\geq \frac{1}{|X|-1} \left( \left( 1 - \frac{1}{|X|} \right) \sum_{i=1}^{|X|} \sigma_i^2 \right) = \frac{\sum_{i=1}^{|X|} \sigma_i^2}{|X|}$$

Therefore, $s^2$ is at least as larger as the average variance $\frac{\sum_{i=1}^{|X|} \sigma_i^2}{|X|}$.

$\square$

### F.2 Proof of the Lyapunov condition

We next prove that $\hat{p}_1, \cdots, \hat{p}_{|X|}$ satisfy Lyapunov condition (Definition F.1).

**Lemma F.4** (Boundedness). *The random variables $\hat{p}_1, \hat{p}_2, \ldots, \hat{p}_{|X|}$ have bounded third moments.*

*Proof.* Observe that the estimated global TPR $= \frac{1}{|X|} \sum_{i=1}^{n} \text{TPR}_i$ and FPR $= \frac{1}{|X|} \sum_{i=1}^{|X|} \text{FPR}_i$ are constants over the random trials with respect to the membership identification algorithm and $x_i$, and by the assumption that TPR $\neq$ FPR (as stipulated in Equation (4)), we prove that $\hat{p}_i$ is upper-bounded and lower-bounded by constant. Specifically,

$$\hat{p}_i = \begin{cases} a = \frac{-\text{FPR}}{\text{TPR}-\text{FPR}} & \text{when } \hat{m}_i = 0, \\ b = \frac{1-\text{FPR}}{\text{TPR}-\text{FPR}} & \text{when } \hat{m}_i = 1. \end{cases}$$

Since $\hat{p}_i$ is bounded, i.e., $|\hat{p}_i - \mu_i| \leq C$ for some constant $C$, all moments are automatically bounded. Specifically, the third-order moments $\mathbb{E}[|\hat{p}_i - \mu_i|^3] \leq C^3$. $\square$

**Lemma F.5** (Limit Property). *The sequence $\hat{p}_1, \hat{p}_2, \ldots, \hat{p}_{|X|}$ satisfies the following:*

$$\lim_{|X| \to \infty} \frac{1}{s_{|X|}^{2+\delta}} \sum_{i=1}^{|X|} \mathbb{E}\left[ |\hat{p}_i - \mu_i|^{2+\delta} \right] = 0 \text{ for } \delta = 1$$

*where $\mu_i = \mathbb{E}[\hat{p}_i] = \gamma_i$ is the mean of $\hat{p}_i$, $\sigma_i^2 = Var(\hat{p}_i)$ is the variance of $\hat{p}_i$, and $s_{|X|}^2 = \sum_{i=1}^{|X|} \sigma_i^2$ is the sum of the variances.*

*Proof.* Assume $\delta = 1$ for simplicity (a common choice). Denote $M_i$ as the third moments of $\hat{p}_i$, i.e., $M_i = \mathbb{E}[|\hat{p}_i - \mu_i|^3]$. From Lemma F.4, we have that there exists $C_i$ such that $M_i \leq C_i$ for all $i$. Without loss of generality, we can assume $C_1 = C_2 = \ldots = C_{|X|} = C$. (A simple choice is to let $C = \max(C_1, C_2, \ldots, C_{|X|})$.) Then, we have

$$\frac{1}{s_{|X|}^3} \sum_{i=1}^{|X|} M_i \leq \frac{1}{s_{|X|}^3} \sum_{i=1}^{|X|} C = \frac{C|X|}{s_{|X|}^3}.$$

Since $s^2_{|X|}$ grows linearly with $|X|$, we have $s^2_{|X|} = O(|X|)$. Therefore, from the algebraic properties of big-O notation, we have $s_{|X|} = O(\sqrt{|X|})$. Thus,

$$\frac{C|X|}{s^3_{|X|}} = \frac{C|X|}{(O(\sqrt{|X|}))^3} = \frac{C|X|}{O((\sqrt{|X|})^3)} = \frac{C|X|}{O(|X|^{3/2})} = \frac{C}{O(\sqrt{|X|})} \to 0 \text{ as } |X| \to \infty.$$

$\square$

Since the independent random variables $\hat{p}_1, \hat{p}_2, \ldots, \hat{p}_{|X|}$ satisfy $\mathbb{E}[\hat{p}_i] = \gamma_i < \infty$ and $\mathbb{E}[\hat{p}_i^2] - \gamma_i^2 < \infty$ for any $i = 1, 2, \ldots, |X|$ ( Lemma F.4), and have the limit property when $\delta = 1$ ( Equation (20)) ( Lemma F.5), we finish the proof that $\{X_i\}_{i=1}^n$ satisfies Lyapunov condition. Therefore, by applying the Lyapunov CLT Theorem F.2, the sum $\frac{\sum_{i=1}^{|X|} \hat{p}_i}{s_n}$ converges in distribution to a normal distribution as $|X|$ increases. This ensures that the confidence interval derived remains valid even when the individual $\hat{p}_i$ are not identically distributed but are independent.

## G   PROOF OF THE SAMPLING ERROR OF DATASET-LEVEL DEBIASING

Under the mild assumption in Appendix F that membership predictions on different target data records are independent (experimentally validated in Figure 4 that $MIA(\theta_j, x_i)$ and $MIA(\theta_j, x_{i'})$ are weakly correlated random variables for any $i \neq i'$, where the randomness is over $\theta_j$ and the randomness of the MIA algorithm), we next provide the formal proof:

Observe that Equation (6) and Equation (7) are empirical average of the membership prediction $MIA(\theta_j, x_i)$ over reference models $\theta_j, j = 1, \cdots, N$ and target data records $x_i, i = 1, \cdots, |X|$, and that the reference models are independently trained on random subsets of the target dataset. Then under the independent assumption and by the additive property of variance for independent random variables, we compute the variance of Equation (6) and Equation (7) as follows.

$$\text{Var}[Equation\ (6)] = \frac{1}{N \cdot |X|} \text{Var}\left[\mathbf{1}_{MIA(\theta\_j, x\_i)=1 \text{ and } x\_i \notin \theta\_j}\right]$$

$$\text{Var}[Equation\ (7)] = \frac{1}{N \cdot |X|} \text{Var}\left[\mathbf{1}_{MIA(\theta\_j, x\_i)=1 \text{ and } x\_i \in \theta\_j}\right]$$

where $\theta_j \xleftarrow{uniform} \{\theta_1, \cdots, \theta_N\}$ and $x_i \xleftarrow{uniform} \{x_1, \cdots, x_{|X|}\}$. Thus the sampling error of estimates in Equation (6) and Equation (7) is asymptotically upper bounded by $O\left(\frac{1}{\sqrt{N \cdot |X|}}\right)$.

## H EXPERIMENT DETAILS AND ADDITIONAL RESULTS

### H.1 EXPERIMENT SETTINGS

**Evaluation setting:** We consider an evaluation granularity of 5%, which is small enough for relatively accurate and meaningful dataset usage inference (DUI). To measure the DUI performance, we compute mean absolute error $\mathbb{E}[|p - \hat{p}|]$ over 30 random trials for each proportion $p \in \{0.00, 0.05, \ldots, 1.00\}$, and then evaluate the maximum error among all proportions, denoted as $\max_p \mathbb{E}[|p - \hat{p}|]$. In each trial, we train the target model $\theta$ on a freshly sampled $p$ proportion of the protected dataset $X$ (using uniformly random sampling or special sampling), combined with randomly sampled remaining training data from a data population pool (that does not overlap with $X$). The size of $X$ in the experiments presented in Section 5.3 is 500, to maintain a challenging and practical relative size. (In practice, the model is usually trained on a mixture of datasets, and the overall size of these additional datasets is typically much larger than the target dataset.) We then test the performance under size 5,000 and 25,000 and show in Section 5.4 that dataset usage estimation error decreases as the dataset size increases.

**Model training details:** We consider three model architectures: a standard 5-layer fully connected network (FC-5), a wide ResNet with width 2 (WRN28-2) (Zagoruyko & Komodakis, 2016), and ResNet-34 (He et al., 2016). We use three benchmark datasets: CIFAR-10 (Krizhevsky et al., 2009), CIFAR-100 (Krizhevsky et al., 2009), and a scaled-down version of ImageNet (Tiny-ImageNet) (Le & Yang, 2015). We sample the protected dataset $X$ from the training set of each benchmark dataset and use the remaining training set as the population pool.

For target models, we train them on a freshly sampled random $p$ proportion of the protected dataset $X$, combined with randomly sampled remaining training data from the population pool (that does not overlap with $X$), such that the complete training dataset size for the target model is always half of the training set of the benchmark dataset. For example, the CIFAR-10 training set consists of 50,000 images. Therefore, the training set size for the target model is 25,000.

For the reference models used for our method and MIA baselines, we train each one on half of $X$ combined with remaining data sampled from the population pool, ensuring the complete training set of the reference model has the same size as the target models.

For all target models and reference models, we trained them to achieve the following test accuracies: for CIFAR-10, 53% for FC-5, and 92% for WRN28-2; for CIFAR-100, 68% for WRN28-2, and 65% for ResNet-34; for Tiny-ImageNet, 58% for WRN28-2 and 54% for ResNet-34.

### H.2 IMPLEMENTATION DETAILS OF BASELINES

**MLE baselines** The MLE baselines require to estimate the distribution for the observation of models trained on each proportion of $X$. To approximate the distribution of $o|p$, empirically, we train $N$ reference models on freshly sampled random $p$-fraction of the target dataset $X$ combined with remaining data sampled from the population pool for each of the 21 possible proportion values (i.e., $p = 0.00, 0.05, \ldots, 1.00$), employing the same algorithm $\mathcal{T}$ that was used for training the target models (in total $21N$ reference models). The complete training set size of the reference model is the same as the reference models used for our method. Regarding the two choices of observation $o$ in the MLE baselines:

- **Joint logits of all records in target dataset (high-dimensional):** Building on prior research (Carlini et al., 2022) which demonstrated that scaled logits for individual data points can serve as effective observations and empirically follow a Gaussian distribution, we naturally extend this approach to a dataset-wide scale. We concatenate these point-wise logits into high-dimensional joint logits and model their distribution using a multivariate Gaussian.

$$\Pr(o|p = q) = \Pr(\text{logit}(X; \theta)|\mathcal{N}(\boldsymbol{\mu_q}, \boldsymbol{\Sigma_q})),$$

  where $\text{logit}(X; \theta)$ is the the logits of model $\theta$ on all records in $X$, and $\mu_q$ and $\boldsymbol{\Sigma_q}$ are its empirical mean and covariance matrix across $N$ reference models trained with $q$ proportion of $X$.
- **Averaged logits of all records in the target dataset (one-dimensional):** To reduce the dimensionality of the observation (and thus enable more accurate empirical mean and covariance estimation), we additionally consider the average of the logits over the dataset $X$, following the approach

in Maini et al. (2021). The likelihood is then modelled as the following one-dimensional Gaussian distribution.

$$\Pr(o|p = q) = \Pr\left(\frac{1}{|X|}\sum_{x \in X}\text{logit}(x;\theta)\Big|\mathcal{N}(\mu_q, \sigma_q^2)\right),$$

where $\mu_q$ and $\sigma_q$ are the empirical mean and variance of the averaged logits of all records in $X$ across $N$ reference models trained with $q$ proportion of $X$.

The **step-by-step procedure** is as follows:

1. For each $q \in \{0.00, 0.05, \ldots, 1.00\}$:
   (a) Train $N$ reference models on a proportion $q$ of $X$ combined with data randomly sampled from the population pool.
   (b) Query the reference models using $X$ to obtain observations $o$(e.g., the average logits) for all data in $X$ and estimate the Gaussian distribution on these observations.
   (c) Query the target model using $X$ to obtain the target observation $o_{\text{tar}}$.
   (d) Calculate the likelihood of observing $o_{\text{tar}}$ given the distribution for $q$.

2. Select the $q$ with the highest likelihood of observing $o_{\text{tar}}$.

**MIA baselines** To demonstrate the importance of debiasing, we also compare our methods against two intuitive baselines: the direct aggregation of MIA guesses and MIA scores. (Some might think that MIA scores indicate the probability or confidence that a data point is a member, which could be used to estimate dataset usage.) We employ the same $2N$ reference models as our method, with each model trained on half of dataset $X$:

- **MIA Guess:** This involves directly counting MIA membership predictions, where the proportion is calculated as the fraction of member predictions: $\hat{p} = \frac{1}{|X|}\sum_{i=1}^{|X|}\hat{m}_i$. For a fair comparison, we use the same MIA algorithms that were employed in our methods.
- **MIA Score:** This takes the average of MIA confidence scores across all data points: $\hat{p} = \frac{1}{|X|}\sum_{i=1}^{|X|}\hat{c}_i$, where $\hat{c}_i$ is the confidence score of RMIA (i.e., the probability of observing that $i$-th record's likelihood to be a member is greater than randomly sampled population data points).

The step-by-step implementation of the MIA algorithms is provided in Appendix C.2 and Appendix C.3. These implementations follow the standard procedures described in the original papers.

### H.3 FORMING SUBGROUPS BY $TPR_i - FPR_i$

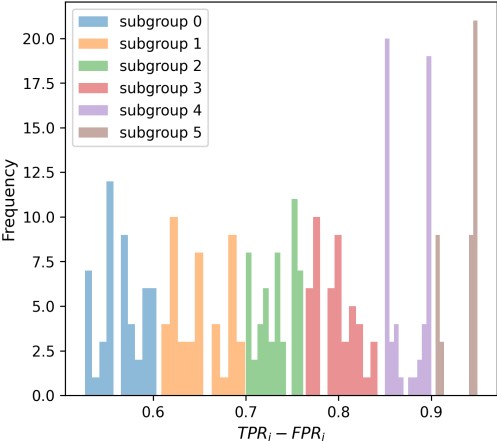

Figure 6: Samples of $TPR_i - FPR_i$ value for 6 subgroups.

## H.4 RESULTS USING DIFFERENT MIA

In the main paper, we present the results of our method using RMIA (Zarifzadeh et al., 2024) due to its superior performance at a limited number of reference models. Additionally, we have implemented another state-of-the-art MIA algorithm, LiRA (Carlini et al., 2022). (Comparisons between RMIA and LiRA can be found in Appendix C.) All results in the main paper, when including the performance of LiRA, are provided in this section.

**Comparison under fixed computation budget** Figure 7 presents the complete version of Figure 2, including the performance of the MIA baseline using LiRA and our method using LiRA. Our methods (red and pink) are clearly more accurate in proportion prediction than all baselines. The comparisons between our method using LiRA and direct predictions from LiRA further confirm the effectiveness of our debiasing method. While our method using RMIA is more robust with a smaller number of reference models compared to using LiRA (analyzed in Appendix C.5), the MIA Baseline (RMIA) and MIA Score (RMIA) show less sensitivity to the number of reference models, possibly due to their inclusion of population data. Overall, they perform better with fewer reference models than the MIA Baseline (LiRA), which is consistent with the results reported in Zarifzadeh et al. (2024).

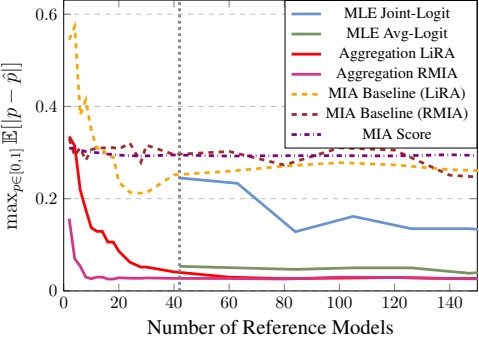

Figure 7: Maximum mean absolute error (MAE) of all methods under an increasing number of reference models. The grey dotted line represents the minimum computational budget required for MLE methods (2 for each proportion). Notably, our method can yield meaningful predictions even under limited computational budget (e.g., a single pair of reference models).

**Confidence interval** Figure 8 adds the confidence interval of our method using LiRA (corresponding to Figure 3 in the main paper). The top figure demonstrates the effectiveness of our methods using both membership identification methods, while the lower figure illustrates that our 95% confidence interval predictions are consistently tight.

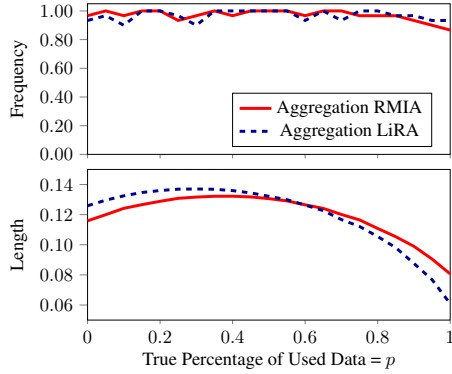

Figure 8: *(Confidence Interval)* **Top:** Frequency of the true proportion $p$ falling within the predicted 95% confidence interval for different $p$. The frequency was computed across 30 interval predictions for each $p$. **Bottom:** Average width of the predicted interval over 30 tests for each $p$.

## H.5 PERFORMANCE OF OUR METHOD IN DETERMINING BINARY DATASET USAGE

While methods restricted to binary predictions under an all-or-none dataset usage scenario cannot ensure consistent predictions for partial utilization (Figure 1), a method providing fine-grained estimates can naturally be reduced to solve the binary problem.

To illustrate this, consider the null hypothesis ($H_0 : s = s' + \tau$) used in prior binary dataset inference literature. For different contexts, $s$ and $s'$ can take the following forms:

1. **Dataset Inference Maini et al. (2021):** $s$ and $s'$ represent the distances to the decision boundary measured on a private dataset and a population dataset, respectively. This hypothesis assumes that if the model was trained on the private dataset, the distance measured on the private dataset would exceed that on the public dataset.

2. **LLM Dataset Inference Maini et al. (2024):** $s$ and $s'$ are the weighted aggregations of 52 MIA scores over the private and population datasets. This hypothesis assumes that merged MIA scores would be significantly higher for members than non-members over enough samples.

3. **Backdoor Watermarks Guo et al. (2023); Li et al. (2023); Tang et al. (2023); Hu et al. (2022); Li et al. (2022b); Wenger et al. (2022); Sablayrolles et al. (2020):** $s$ and $s'$ are the confidence score on the target label given backdoored inputs and given clean inputs. This hypothesis assumes that a model trained on a poisoned dataset (if successfully backdoored) will assign higher confidence to the target class when triggered, but not for clean inputs.

For DUCI, a straightforward simplification to the dataset inference problem can be made by set $s = \hat{p}$, $s' = 0$, and $\tau$ serves as a threshold, which may vary depending on the data type. Table 6 report the performance of our method adapted for solving the binary dataset inference problem.

Table 6: Comparison of p-values between DUCI and binary dataset usage algorithms for determining whether a dataset $X$ (size 500) has been used. The complete training dataset of the target model has a size of 25,000. For p-values, a smaller value for **Dataset Used** is better, while a larger value for **Dataset Not Used** is better.

| Methods | p-value | |
|---|---|---|
| | Dataset Used ↓ | Dataset Not Used ↑ |
| Backdoor Watermark (poison 30% of $X$) | $7.10 \times 10^{-5}$ | 0.334 |
| Backdoor Watermark (poison 100% of $X$) | $\mathbf{6.18 \times 10^{-54}}$ | **1.000** |
| Dataset Inference | $7.27 \times 10^{-10}$ | 0.937 |
| Ours | $\mathbf{3.15 \times 10^{-51}}$ | **1.000** |

Consistent with the performance shown in Figure 1, all methods can perfectly solve the binary dataset usage problem when the significance level is set to common thresholds such as 0.05 or 0.01. This challenge becomes especially critical and significantly impacts performance when the dataset is not fully sampled or when the protected dataset's relative size is small. Our method performs exceptionally well, achieving comparable results to backdoor watermarking when the entire dataset is poisoned. In principle, the performance of Dataset Inference should be close to our method; however, the slight drop in performance may be attributed to the choice of signal, as the loss-based score is less distinguishable in distribution than the likelihood ratio-based score. We did not compare with (Maini et al., 2024) as combining multiple MIAs is orthogonal to our approach. Our method can debias any number of MIAs using the same reference models without retraining, with combination possible after debiasing if needed.

Finally, it is important to note that directly comparing the reported error of DUCI at $p = 0$ and $p = 1$ to that of dataset inference is inherently unfair. DUCI predicts a continuous value, whereas dataset inference is a simple binary classification task.

# I  ON (THE POSSIBILITY OF) IMPROVING DATASET USAGE INFERENCE VIA PAIR-WISE "CORRELATION"

Our method in Section 4 treat individual records in the target dataset separately, and thus implicitly require "independence" among records to perform well. **Will the ignorance of "correlations" between records predictions make our method sub-optimal for the problem of dataset usage inference? If the "correlations" between records predictions exists, how can we debias membership predictions?** In this section, we explore the room for improving our aggregation method via exploiting correlations among different records in the target dataset. We first show that under natural datasets (without manually crafted data), the room for improvement is small as the correlation between records prediction is almost zero. However, in Appendix I.2, we demonstrate that with a dataset containing specially designed canaries (i.e., correlation exists), our method can be extended to second-order debiasing given pairwise membership predictions, enabling more accurate dataset usage inference.

## I.1  ON THE POSSIBILITY OF COVARIANCE REDUCTION

For the sake of understanding the role of correlation, let us now focus on the simplest scenario of inferring the usage of a dataset with two records. Under such settings, the mean squared error (MSE) for our proportion estimator $\hat{p} = \frac{1}{|X|} \sum_{i=1}^{|X|} \hat{p}_i$ in Section 3.1 is given by:

$$\text{MSE}(\hat{p}) = \mathbb{E}\left[\left(\frac{1}{2}(\hat{p}_1 + \hat{p}_2) - p\right)^2\right] \tag{33}$$

$$= \frac{1}{4}\text{Var}(\hat{p}_1) + \frac{1}{4}\text{Var}(\hat{p}_2) + \frac{1}{2}\text{Cov}(\hat{p}_1, \hat{p}_2) \tag{34}$$

where the last equality is by the unbiasedness property of $\hat{p}_1$ and $\hat{p}_2$. This naturally motivates us to ask: *Can we reduce the correlation (i.e., covariance term defined in Equation* (34)*) between data pairs in the target dataset, thereby reducing the error of dataset usage inference estimator?* To answer this question, we evaluate the magnitude of correlations among data records in the target dataset, and plot the histogram in Figure 4. We observe that a majority of data pairings incur near-zero covariance values. This is in line with prior works Pillutla et al. (2023) that observe small correlation between membership guesses on different data records. Only a small number of outliers (less than 0.01%) show positive covariance values larger than $0.05$ – these pairs sometimes consist of records very similar to each other (examples are provided in Figure 10). These trends imply that the room for improving our aggregation methods via reducing covariance between data pairs is *generally very small*. It is an intriguing open problem as to whether our method could be further improved if the data owner is allowed to enforce high correlations among different data records, e.g., by modifying the target dataset.

## I.2  ON THE POSSIBILITY OF PAIR-WISE BIAS REDUCTION (SECOND-ORDER DEBIASING)

*If correlations between membership predictions exist (i.e., $\mathbb{E}[\hat{p}_i\hat{p}_j] - \mathbb{E}[\hat{p}_i]\mathbb{E}[\hat{p}_j] \neq 0$), is it still possible to aggregate group-level statistics unbiasedly to predict dataset usage?* In this section, we provide an example of second-order debiasing (when all records in the target dataset are uniformly randomly sampled, i.e, $\gamma_i = p$ for any record $i$ when the ground-truth proportion is $p$).

**Debiasing pair-wise membership guesses to estimate dataset usage proportion** Suppose a membership identification method is able to predict pair-wise memberships $\hat{m}_i\hat{m}_j$ for any pair of points $(x_i, x_j)$ where $i, j \in 1, 2, \ldots, |X|$. We now apply it to randomly sampled data record pairs from the target dataset. For each data pair $(x_i, x_j)$, denote $m_i$ and $m_j$ as their ground-truth memberships, and denote $\hat{m}_i$ and $\hat{m}_j$ as their pair-wise membership guesses (by definition $m_i, m_j, \hat{m}_i, \hat{m}_j \in \{0, 1\}$). Let $P(\hat{m}_i\hat{m}_j = 00)$, $P(\hat{m}_i\hat{m}_j = 01)$, $P(\hat{m}_i\hat{m}_j = 10)$ and $P(\hat{m}_i\hat{m}_j = 00)$ be the frequency of each pair-wise guesses.

To estimate the dataset usage proportion, we need to perform debiasing operations (similar to Section 4[10]) on the observed pair-wise membership guesses. For this, we first estimate the following matrix $M$ (i.e., all off-diagonal values in this matrix represent error types, analogous to TPR and FPR) of conditional distribution of the pair-wise membership guesses $\hat{m}_i \hat{m}_j$ given different ground-truth pair-wise membership values $m_i m_j$, on empirically trained *reference models* (of which we know the training datasets).

$$
M = \begin{bmatrix} P(00|00) & P(00|01) & P(00|10) & P(00|11) \\ P(01|00) & P(01|01) & \cdots & \cdots \\ \vdots & \vdots & \ddots & \vdots \\ P(11|00) & \cdots & \cdots & P(11|11) \end{bmatrix},
$$

where for brevity we denoted $P(a|b)$ to be $P(\hat{m}_i \hat{m}_j = a | m_i m_j = b)$, and the randomness is over the training algorithm and the random data pairs.

Therefore, by the total probability, we have

$$
M \cdot \begin{bmatrix} P(m_i m_j = 00) \\ P(m_i m_j = 01) \\ P(m_i m_j = 10) \\ P(m_i m_j = 11) \end{bmatrix} = \begin{bmatrix} P(\hat{m}_i \hat{m}_j = 00) \\ P(\hat{m}_i \hat{m}_j = 01) \\ P(\hat{m}_i \hat{m}_j = 10) \\ P(\hat{m}_i \hat{m}_j = 11) \end{bmatrix} \tag{35}
$$

To estimate the dataset usage proportion, one only need to observe that when a randomly sampled $q$-proportion of the target dataset is used for training, the ground-truth pair-wise membership values follow the following distribution: $P(m_i m_j = 11) = q^2$, $P(m_i m_j = 01) = P(m_i m_j = 10) = (1-q)q$, and $P(m_i m_j = 00) = (1-q)^2$. By plugging them to Equation (35), we can estimate the dataset usage proportion by finding a $q$ that enables the closest observations to the empirically observed pair-wise membership guesses, as follows.

$$
\hat{p} = \arg\min_q \mathbb{E} \left[ \left\| M \begin{bmatrix} (1-q)^2 \\ (1-q)q \\ q(1-q) \\ q^2 \end{bmatrix} - \begin{bmatrix} P(\hat{m}_i \hat{m}_j = 00) \\ P(\hat{m}_i \hat{m}_j = 01) \\ P(\hat{m}_i \hat{m}_j = 10) \\ P(\hat{m}_i \hat{m}_j = 11) \end{bmatrix} \right\|_2^2 \right]
$$

### I.2.1 EXAMPLE OF PAIR-WISE DEBIASING GIVEN LIKELIHOOD-BASED MEMBERSHIP PREDICTION

We begin by selecting the most basic pair-wise statistic—pair-wise likelihood (analogous to loss and considered the weakest MIA signal)—as the basis for generating pair-wise membership predictions. As baselines, we include likelihood-ratio-based methods (a stronger likelihood-based signal), per-point MIA (LiRA Carlini et al. (2022)), and the best signal (RMIA). Our results demonstrate that, under second-order debiasing, even the weakest signal performs comparably to the aggregation of membership predictions using the best first-order signal.

**Pair-wise membership inference**   The idea of pair-wise membership inference is to design a test that can simultaneously guess the membership status of a pair of points $(x_i, x_j)$ for any $i, j \in 1, 2, \ldots, |X|$. We can use the general maximum likelihood estimation to design a pair-wise test, as follows.

$$
\hat{m}_i \hat{m}_j = \arg \max_{s \in \{00, 01, 10, 11\}} P(o | m_i m_j = s) \tag{36}
$$

where we consider the observation $o$ as averaged logits of the data pair $(x_i, x_j)$ and model the distribution of this observation $o$ when $m_i m_j = s$ for each $s \in \{00, 01, 10, 11\}$ as a multivariate Gaussian, similarly as Equation (10).

---

[10]The second-order debiasing is a natural extension of straightforward debiasing since under uniformly random sampling, method in Section 4 is solving $\hat{p} = \arg\min_q \mathbb{E} \left[ \left\| M \begin{bmatrix} (1-q) \\ q \end{bmatrix} - \begin{bmatrix} P(\hat{m}_i = 0) \\ P(\hat{m}_i = 1) \end{bmatrix} \right\|_2^2 \right]$ in principle.

**Performance of the second-order debiasing** We refer to our method presented in the main paper, which is based on the aggregation of debiased individual membership guesses, as the aggregating individual statistics method (Agg Individual) to distinguish it from the pair-wise test. In Figure 9, we observe that the pair-wise test enables better performance (smaller bias) for proportion estimation compared to the Agg Individual method using LiRA. This suggests (1) the second-order debiasing is accurate, and (2) that pair-wise statistics (naive likelihood value) may capture more information than the corresponding individual statistics (stronger likelihood ratio).

However, we observe that the pair-wise test does not outperform the RMIA-based Agg Individual method (which is stronger than the LiRA-based method). We hypothesize that this is because the score (likelihood based rank score) used in RMIA is much stronger than the score used in LiRA and our pair-wise test. The RMIA score captures more data information, bridging the gap between first-order and second-order statistics. It is an intriguing open question whether we can design more powerful pair-wise tests using stronger MIA scores like those used in RMIA.[11]

Additionally, it is important to note that the current pair-wise test requires much higher computational cost than the aggregation of individual statistics due to the computation of MLE for each sampled data pair. The computational cost for inference of the pair-wise test is $O(|X|^2)$, while our Agg Individual method is $O(|X|)$. Therefore, identifying ways to reduce this cost could be another direction for future research.

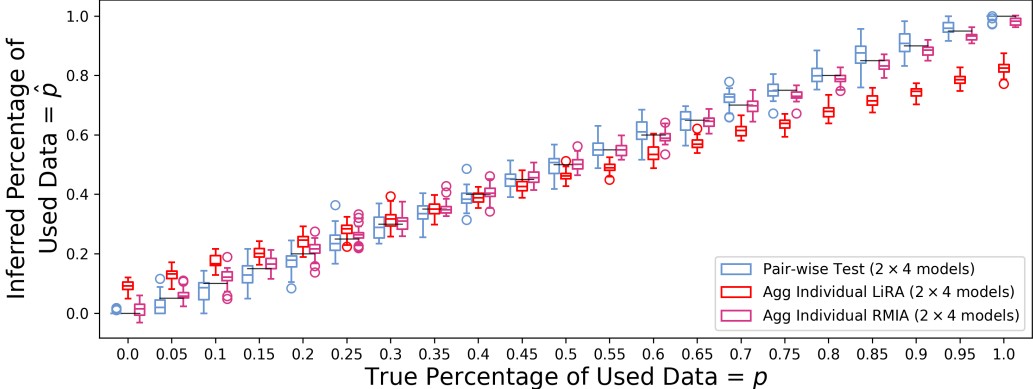

Figure 9: Comparison between the dataset usage inference using pair-wise statistics and individual statistics (red and pink) when only 4 pairs of reference models are used. For the pair-wise test, we follow Appendix I.2 and aggregate the debiased pair-wise membership guesses over 20000 randomly sampled data pairs in the target dataset.

---

[11]There is no straightforward method for adapting RMIA scores for pair-wise testing.

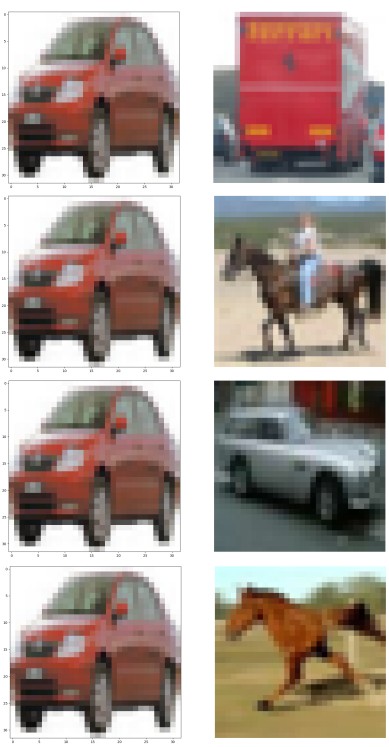

Figure 10: Examples of pairs with large covariance.

## J CASE STUDY: QUANTIFYING BOOK COPYRIGHT INFRINGEMENT

The issue of copyright infringement in books can be considered a specific case of data usage inference: given a model that likely infringes on book copyrights, the author or publishing house would want to determine the extent to which their content is used. We conducted our experiment using pre-trained GPT-2 (Radford et al., 2019). For a rigorous setting, we selected books from the BookMIA dataset constructed by Shi et al. (2023). This dataset contains 50 new books with first editions in 2023, which could not have been included in the pre-training data of GPT-2, and 50 old books—known to be memorized by ChatGPT—whose memorization status by GPT-2 is uncertain. To prevent data contamination and distribution shifts between members and non-members Zhang et al. (2024), we only use the 50 new books from the BookMIA dataset Shi et al. (2023) for evaluation. These books are partitioned into a protected pool of 30 books and a population pool of 20 books, with each sentence treated as a separate entry.

**Target Model:** We consider six different infringement proportions $p \in \{0.0, 0.2, \ldots, 1.0\}$ and trained $m = 4$ target models, each fine-tuned on different proportions of protected books.

Specifically, for each of the $m$ target models, we randomly partition the protected pool into six disjoint sets of books $S_0, S_{0.2}, S_{0.4}, S_{0.6}, S_{0.8}$, and $S_{1.0}$, where each $S_p$ contains $n = 5$ target books. The fine-tuning dataset for each target model is then constructed as follows:

1. **Protected books:** For each protected book $X \in S_p$ for $p \in \{0.0, 0.2, 0.4, 0.6, 0.8\}$, we select the *beginning* $p$ proportion of the sentences from the protected book $X$.

2. **Outsourced books:** We add another $n = 5$ randomly sampled books from the 50 old books (as the remaining dataset from unknown sources).

Therefore, for each of these $m = 4$ target models, we have $n = 5$ target books with $p$ proportion of content included in the training for each possible proportion value $p \in \{0.0, 0.2, 0.4, 0.6, 0.8, 1.0\}$. This setup results in a total of 20 trials for each $p$.

Note that in this experiment, the used sentences are *sequentially* chosen rather than uniformly randomly chosen as in Section 5.1, to better reflect practical scenarios where text continuity in a book is preserved.

**Reference model:** We train two reference models, each with a fine-tuning set that includes two parts:

1. **Protected books:** Each model is trained on a random half of each book in the protected pool of 30 books. More specifically, both models are trained on all 30 books, but each model uses only half amount of the sentences from each book without overlap.

2. **Population books:** Each model also includes half of the sentences from $2n = 10$ randomly selected books from the population pool of 20 books as the remaining data.

This ensures that 1) for each sentence used in training the target model, one reference model is fine-tuned on it while the other is not, and 2) among the remaining data (exclusive the target books), we can select sentences that are present in only one of the reference models as the *population data* (used for computing the MIA scores only).

All target and references models incur training loss of around 3.2 and test loss of around 4.0, which is close to the loss reported by Prat (2023). The time for fine-tuning a single model (both target and reference models) for 3 epochs on a single NVIDIA GeForce RTX 3090 GPU is approximately 20 minutes.

**Experiment setting** We use the RMIA without population data as the underlying MIA algorithm (following the implementations in Meeus et al. (2025); Zarifzadeh et al. (2024)) for our methods and MIA Guess baseline due to the significant costs of population data inference, and use the RMIA with the above-mentioned population data only for the MIA Score baseline. We select the optimal MIA threshold as described in Appendix C.4 on reference models.[12] We consider MIA Guess and MIA Score as our baselines:

---

[12] Specifically, among the two reference models, the one used as the target model for threshold determination or debiasing will not be used as the attack model.

1. **MIA Guess**: For each book $X$ with sentences $\{c_i\}_{i=1}^{|X|}$, we perform MIA on each chunk $c_i$ to obtain an individual membership prediction $\hat{m}_i$. We then compute the ratio $\hat{p} = \frac{1}{|X|} \sum_{i=1}^{|X|} \hat{m}_i$.

2. **MIA Score**: Similar to MIA Guess, we conduct the same process but, instead of obtaining binary predictions, we directly aggregate the MIA scores as described in Section 5.2.

For our methods, we repeat the debiasing process described in Algorithm 1 given by $\mathbb{E}(\hat{m}|m = 0)$ and $\mathbb{E}(\hat{m}|m = 1)$ estimated on top of the membership guesses provided by the MIA.

The inference time over 30 target books for all methods, including the estimation of $P(\hat{m}|m = 0)$ and $P(\hat{m}|m = 1)$, on a single NVIDIA GeForce RTX 3090 GPU takes 408 seconds.

**Results and analysis** Although MIA on text generation models has been explored since (Song & Shmatikov, 2019), previous research typically conducts MIA by querying individual texts and gathering a single statistic to infer whether the entire collection is part of the training data. A few discussions on book copyright infringement issue in prior works (Radford et al., 2019) also focus on this Yes or NO question, which is just the binary version of our dataset usage inference problem.

Our experiment shows that the naive aggregation of membership inference methods struggle to distinguish between different levels of infringement, particularly when the infringement is minimal. As shown in Table 7, the MIA baseline tends to over-rely on the behavior of the half-half split reference model, leading to large errors when the true usage proportion deviates from $p = 0.5$. In contrast, our method effectively mitigates this issue. For example, MIA Score and MIA Guess baselines incur maximum mean absolute errors of 0.5 and 0.335, respectively, across tested proportions, whereas our method achieves a much lower error of 0.168. Notably, while our estimator is provably unbiased under independent sampling—a condition violated in this correlated, contiguous sampling experiment—it still performs reliably, demonstrating its robustness in estimating the extent of copyright infringement under special sampling conditions.

Interestingly, we also observe generally higher error variance for our method across different proportions on text data compared to image data. This is because the variance of our estimator increases as the gap TPR − FPR decreases. Just for illustration, assuming each record is independently sampled with equal probability, the variance of the estimator is

$$\text{Var}[\hat{p}] = \frac{\text{Var}[\hat{m}]}{(\text{TPR} - \text{FPR})^2} = \frac{p(1-p)}{(\text{TPR} - \text{FPR})^2} \propto \frac{1}{(\text{TPR} - \text{FPR})^2}.$$

Given that memberships on text data are typically less distinguishable due to larger training corpora and more severe model forgetting, it tends to have a smaller TPR − FPR, and thus higher estimation variance, as reflected in the larger MAE variance.

More surprisingly, we observed that RMIA with only reference models is more stable than complete-version of RMIA (with population data). The broad distribution of population data makes RMIA scores less stable across models. Instead, the normalization term computed over reference models plays a key role in improving the effectiveness of debiasing. That's saying, a more powerful attack does not necessarily yield better performance under debiasing. Instead, weaker yet more stable attacks (i.e., those producing comparable scores across models) benefit more from DUCI.

Table 7: Mean Absolute Error $\mathbb{E}[|\hat{p}_i - p|]$ for different values of $p$. The lowest error under each proportion $p$ is highlighted.

| proportion $p$ | MIA Guess | MIA Score | DUCI (Our Method) |
|---|---|---|---|
| 0.0 | 0.3350 | 0.5019 | **0.1684** |
| 0.2 | 0.2396 | 0.4029 | **0.1573** |
| 0.4 | 0.1202 | 0.2381 | **0.0833** |
| 0.6 | 0.0439 | 0.1028 | **0.0316** |
| 0.8 | 0.1286 | 0.0720 | **0.0453** |
| 1.0 | 0.2448 | 0.2189 | **0.0568** |
| $\max_{p \in [0,1]} \mathbb{E}[|\hat{p}_i - p|]$ | 0.3350 | 0.5019 | **0.1684** |

