# OpenReview forum: "How much of my dataset did you use? Quantitative Data Usage Inference in Machine Learning"
_ICLR.cc/2025/Conference — ICLR 2025 Oral_

### Official Review · Reviewer_BehN · 2024-11-01

**Soundness:** 3
**Presentation:** 3
**Contribution:** 2
**Rating:** 6
**Confidence:** 2

**Summary:**

This paper proposes an algorithm framework that can figure out whether data points in the data sets are used to train a model. This problem seems interesting, and there are some works on a similar problem called membership inference. The authors propose that instead of predicting each member by {0,1}, a better way is to predict a probability in [0,1] and design an algorithm for DUCI based on this idea. The authors also did many experiments comparing their method with several baselines on errors and confidence intervals.

**Strengths:**

This paper studies an interesting problem and proposes a new algorithm to predict a probability in [0,1] instead of {0,1} for the problem. Though the authors proposed a specific algorithm, the technique here can be used for any algorithm that serves the purpose of membership query. This paper is well-written and easy to read. The authors also did experiments thoroughly by comparing with baselines and on different datasets.

**Weaknesses:**

Though this paper has some novelty, the technique here seems to be quite simple and straightforward. It is hard for me to say this paper has good contribution confidently.

**Questions:**

The authors mention two possible improvements in Appendix G. I think this paper would be really strong if the authors could be more concrete on how to apply one of the ideas to their algorithm and show some results.

---

> ### Author Response · Authors · 2024-11-22
> **To Reviewer BehN**
>
> > 1. The authors mention two possible improvements in Appendix G. I think this paper would be really strong if the authors could be more concrete on how to apply one of the ideas to their algorithm and show some results.
>
> We sincerely appreciate your efforts in reviewing the improved methods in the Appendix! In response to your question, we have updated the details of the second-order debiasing methods in Appendix H.2 in the [revised paper](https://openreview.net/pdf?id=EUSkm2sVJ6). However, we maintain the simple first-order dataset-level debiasing method in the main paper, ***as one of our primary focus is to show that the key to solving DUCI problem (via existing MI techniques) lies in the debiasing concept itself, independent of the specific debiasing design. We believe the conciseness and straightforwardness of main methods are essential for effectively conveying this concept.*** Additionally, as we have demonstrated, ***this simple debiasing approach can serve as a foundation for various refinements***:
>
> First, **the unit of debiasing is adjustable**: As shown in Footnote 1 and Table 2, this method can be extended to subgroup-level debiasing to address special sampling scenarios---a general challenge faced by all baselines and previous dataset inference techniques.
>
> Second, it can be enhanced to **higher-order debiasing** (to membership inference methods can leverage higher-order statistics) to capture the correlation between the membership prediction between records: We present an example of second-order debiasing method in the Appendix H.2 for cases where records in the target dataset are uniformly randomly sampled. However, this example method faces an increased algorithmic complexity (from $O(|X|)$ for first-order debiasing to $O(|X|^2)$ for second-order). Developing more efficient higher-order DUCI methods remains an intriguing direction for future work.

---

> > ### Comment · Reviewer_BehN · 2024-11-23
> >
> > Thank you for your clarifications. I will keep my score.

---

> > > ### Author Response · Authors · 2024-11-24
> > >
> > > Thank you once again for your time and effort in reviewing our paper!

---

### Official Review · Reviewer_Tzw7 · 2024-11-03

**Soundness:** 3
**Presentation:** 3
**Contribution:** 4
**Rating:** 8
**Confidence:** 3

**Summary:**

Given a dataset, this paper presents an algorithm (DUCI) which estimates the proportion of that dataset used in the training of a model. The algorithm estimates the false positive rate (FPR) and true positive rate (TPR) of the membership inference guess across the entire dataset to avoid the accumulation of errors that occurs when estimating the FPR and TPR of each individual in the dataset. They conduct experiments to compare the performance of their algorithm (DUCI) against traditional membership inference baselines and an idealized, computationally inefficient MLE baseline. They also analyze the performance of DUCI and membership inference baselines under special sampling conditions and varying dataset sizes.

**Strengths:**

The paper is well motivated and addresses a gap in the literature by taking a fine grained approach to the data usage problem.

The proposed approach (DUCI) is significantly more computationally efficient compared to previous approaches.

**Weaknesses:**

I don't see any major weaknesses. It would be nice to show a comparison between DUCI and SOTA “binary” data usage algorithms for the specific case for when p=1 and p=0 to demonstrate that DUCI still has comparable performance to “binary” data usage algorithms in these specific cases.

**Questions:**

If TPR=FPR, how does this affect the debiasing results?

---

> ### Author Response · Authors · 2024-11-22
> **To Reviewer Tzw7**
>
> We thank the reviewer for the positive feedbacks and helpful suggestions for further improvement!
>
> > I don't see any major weaknesses. It would be nice to show a comparison between DUCI and SOTA “binary” data usage algorithms for the specific case for when p=1 and p=0 to demonstrate that DUCI still has comparable performance to “binary” data usage algorithms in these specific cases.
>
> Thanks for the suggestion! Below is the formulation of using DUCI for the binary dataset inference problem, with results demonstrating the effectiveness of our methods in binary scenarios. We have also included this comparison in Appendix G.5 of the [revised paper](https://openreview.net/pdf?id=EUSkm2sVJ6).
>
> ***Hypothesis Testing Formulation for DUCI in Binary Dataset Inference Problems***
>
> Prior binary dataset inference literature consider the binary hypothesis test to solve the problem, where the null hypothesis has a general form $H_0: s = s' + \tau$ (i.e., the target model is not trained on the protected dataset). For different methods, $s$ and $s'$ can take the following forms:
> 1. **Dataset Inference [1]**: $s$ and $s'$ represent the distances to the decision boundary measured on a private dataset and a population dataset, respectively. This hypothesis assumes that if the model was trained on the private dataset, the distance measured on the private dataset would exceed that on the public dataset.
> 2. **LLM Dataset Inference [2]**: $s$ and $s'$ are the weighted aggregations of 52 MIA scores over the private and population datasets. This hypothesis assumes that merged MIA scores would be significantly higher for members than non-members over enough samples.
> 3. **Backdoor Watermarks [3]**: $s$ and $s'$ are the confidence score on the target label given backdoored inputs and given clean inputs. This hypothesis assumes that a model trained on a poisoned dataset (if successfully backdoored) will assign higher confidence to the target class when triggered, but not for clean inputs.
>
> Regarding DUCI, a straightforward simplification from DUCI to the dataset inference problem can be made by set $s = \hat{p}$, $s' = 0$, and $\tau$ serves as a hyperparameter, which may vary depending on the data type. Below, we report the performance of our method adapted for solving the binary dataset inference problem.
>
> **Table:** Comparison of p-values between DUCI and binary dataset usage algorithms for determining whether a dataset $X$ (size 500) has been used. The complete training dataset of the target model has a size of 25,000. For p-values, a smaller value for **Dataset Used** is better, while a larger value for **Dataset Not Used** is better.
>
> | **Methods**                           | **p-value (Dataset Used ↓)** | **p-value (Dataset Not Used ↑)** |
> |---------------------------------------|------------------------------|-----------------------------------|
> | Backdoor Watermark (poison 30% of $X$) | $7.10 \times 10^{-5}$      | 0.334                            |
> | Backdoor Watermark (poison 100% of $X$) | $\mathbf{6.18 \times 10^{-54}}$ | **1.000**                        |
> | Dataset Inference                     | $7.27 \times 10^{-10}$     | 0.937                            |
> | Ours                                  | $\mathbf{3.15 \times 10^{-51}}$ | **1.000**                        |
>
> Consistent with the performance shown in Figure 1, all methods can perfectly solve the binary dataset usage problem when the significance level is set to common thresholds such as 0.05 or 0.01. **For backdoor watermark methods, the main challenge lies in the successful injection of backdoor when the dataset is not fully sampled or when the protected dataset's relative size is small. This will significantly impact performance, e.g., poisoning even 30% of X performs poorly when $|X|$ is small.
> Our method performs exceptionally well, achieving comparable results to backdoor watermarking when the entire dataset is poisoned.** In principle, the performance of Dataset Inference should be close to our method; however, the slight drop in performance may be attributed to the choice of signal, as the loss-based score is less distinguishable in distribution than the likelihood ratio-based score.
>
> *[1] Maini, P., Yaghini, M., & Papernot, N. (2021). Dataset inference: Ownership resolution in machine learning.*
>
> *[2] Maini, P., Jia, H., Papernot, N., & Dziedzic, A. (2024). LLM Dataset Inference: Did you train on my dataset?.*
>
> *[3] Li, Y., Zhu, M., Yang, X., Jiang, Y., Wei, T., & Xia, S. T. (2023). Black-box dataset ownership verification via backdoor watermarking.*

---

> > ### Author Response · Authors · 2024-11-22
> >
> > > 2. If TPR=FPR, how does this affect the debiasing results?
> >
> > Good question. It would result in a zero denominator in Equation (6), so we actually assume $TPR \neq FPR$ in Line 205. As discussed in Lines 206–207, this assumption should be reasonable and achievable for the following reasons:
> >
> > 1. A membership identification method designed to discriminate between member and non-member data points, rather than acting randomly, is unlikely to produce $TPR = FPR$ all the time, especially when the dataset is not random.
> >
> > 2. Since $TPR$ and $FPR$ can be adjusted by varying the threshold, it should always be possible to select a threshold such that $TPR \neq FPR$.

---

### Official Review · Reviewer_H5r7 · 2024-11-04

**Soundness:** 3
**Presentation:** 3
**Contribution:** 3
**Rating:** 8
**Confidence:** 2

**Summary:**

This paper presents a new variation of membership attack: estimating the fraction of data from training set. Unlike normal MIA, which determine individual membership, the proposed task estimate the data usage directly. The proposed algorithm is based on the fact that the  unbiased data usage estimator can be written as a function of FPR and TPR (Eq. 6) of MIA. In other words, the proposed estimator can adapt any existing MIA attack with FPR and TPR evaluation. The experiments demonstrate the effectiveness.

**Strengths:**

This paper introduces a new task in privacy attacks. The main contribution is a practical and scalable data usage estimator that could encourage further research in this area.

**Weaknesses:**

My main concern is that this method requires known training set for estimating FPR and TPR. In practice, the train set is usually private (see following reference Zhang at el 2024).



Zhang, Jie, Debeshee Das, Gautam Kamath, and Florian Tramèr. "Membership Inference Attacks Cannot Prove that a Model Was Trained On Your Data." arXiv preprint arXiv:2409.19798 (2024).

**Questions:**

1. In Figure 3, the length of confidence interval seems to be large compared to the absolute error. Is this true?

2. Would this debiasing method downgrade the test power?

3. Is there any connection with auditing differential privacy?

---

> ### Author Response · Authors · 2024-11-22
> **To Reviewer H5r7**
>
> Thank you for your important and interesting questions. We believe all our discussion will offer valuable insights for future work.
>
> > 1. My main concern is that this method requires known training set for estimating FPR and TPR. In practice, the train set is usually private (see following reference Zhang at el 2024).
>
> Thanks for raising this intriguing question. We are aware of the challenges in designing a non-member scenario for evaluating training data proof, as discussed by Zhang et al., 2024. Three key issues in the current mainstream design of evaluation that related to our task are as follows:
>
> 1. **Distribution shifts**: Non-member data collected based on the model's training cutoff date can introduce distribution shifts, making the evaluation artificially easier.
> 2. **Member uncertainty**: Not all data released before the cutoff are guaranteed to be members since the cutoff date provided by model developers may be inaccurate.
> 3. **Causal relationships**: Using hold-out counterfactual data as non-members can introduce causal dependencies, as the model may have been trained on a related, recently released version of the data.
>
> To ensure reliable evaluation in our book copyright infringement case study, we fine-tune a GPT-2 model on a recently collected (by cutoff date) dataset, ensuring that the original training set has no causal relationship with the new data. Instead of directly treating the entire dataset as non-member data, we fine-tune the model on different proportion of data to create members and non-members. This ensures there is no distribution shift between members and non-members that could be exploited.  Our focus here is to develop effective methods for the DUCI problem and demonstrate their superiority over baselines under a fairly designed evaluation, which we achieved.
>
> ***Regarding the concern about evaluation on large production models:*** we agree that closed-source training pipelines of production models pose problems. However, this issue is independent of method design, and all training data proof evaluations face the same problems. This calls for more open-source (like Pythia [1]) evaluation benchmarks on different model architectures and datasets.
>
> Lastly, regarding the TPR/FPR values used for debiasing in our method: we do not assume access to the target model's training set. Instead, TPR and FPR are estimated on our private dataset (known to the dataset owner) using a reference model. This reference model could be finetuned based on a checkpoint released before the dataset’s creation or even a smaller model with a different architecture and trained on different data [2,3].
>
> *[1] Biderman, S., Schoelkopf, H., Anthony, Q. G., Bradley, H., O’Brien, K., Hallahan, E., ... & Van Der Wal, O. (2023, July). Pythia: A suite for analyzing large language models across training and scaling.*
>
> *[2] Duan, M., Suri, A., Mireshghallah, N., Min, S., Shi, W., Zettlemoyer, L., ... & Hajishirzi, H. (2024). Do membership inference attacks work on large language models?*
>
> *[3] Nicholas Carlini, Florian Tramer, Eric Wallace, Matthew Jagielski, Ariel Herbert-Voss, Kather- ine Lee, Adam Roberts, Tom B Brown, Dawn Song, Ulfar Erlingsson, et al. Extracting Training Data from Large Language Models.*
>
> > 2. In Figure 3, the length of confidence interval seems to be large compared to the absolute error. Is this true?
>
> Thanks for pointing out the potential confusion; we will improve the clarity of the confidence interval (CI) explanation in our paper. First, **the small value of MAE compared to the length of the 95\% CI illustrates that our predictions are highly concentrated around the true value**, as if the ***unbiasedness of $\hat{p}$ is empirically achieved, MAE equals to the standard deviation $\sigma$ of $\hat{p}$***. Second, given that the ***95\% confidence interval is calculated according to Lyapunov CLT*** ($\hat{p}$ follows an approximately Gaussian distribution), ***its length is theoretically equal to $(\mu + 2\sigma) - (\mu - 2\sigma) = 4\sigma$, which is approximately four times the $sigma)$ (MAE)***.
>
> As observed, the maximum length of the 95\% confidence interval (CI) is approximately 0.12, while the maximum absolute error using a single reference model is 0.027 in Table 1—roughly one-quarter of the CI length. This shows that **the CI length is quite small and effectively demonstrates: (1) the empirical unbiasedness of our estimator and (2) the high concentration of our predictions around the true value.**

---

> > ### Author Response · Authors · 2024-11-22
> >
> > > 3. Would this debiasing method downgrade the test power?
> >
> > It would not. If "test power" refers to dataset cardinality inference, our experiments in Tables 1–4 clearly demonstrate that, without the debiasing process, directly aggregating MIA predictions results in significant error. The debiasing process substantially improves the test power for dataset cardinality inference.
> >
> > If "test power" refers to membership inference (MI), the debiasing method does not affect it, as it serves as a post-processing step for membership predictions. That is, if MI is performed by comparing scores to a threshold, the same post-processing can be applied to adjust the threshold, ensuring no degradation in MIA performance. Moreover, intuitively, in scenarios where a single threshold is applied across all points, directly debiasing individual scores acts as a form of calibration, enabling consistent thresholding and potentially improving test power. However, since our primary goal is not to design best MIA, we leave such potential application for future work. In our design, the debiasing process is independent of the chosen MIA method, and the membership predictions remain unchanged within our framework.
> >
> > > 4. Is there any connection with auditing differential privacy?
> >
> > Yes! **DP auditing via DUCI is theoretically feasible. When the training algorithm satisfies differential privacy ($\varepsilon$-DP), provable upper bounds for the error of DUCI can be established via standard packing argument [1].** Loosely speaking, let the ground truth membership probability for record $i$ in the training dataset be $p_i$. By definition, datasets sampled under probabilities $(p_1, \cdots, p_n)$ and $(p_1 \pm \frac{1}{n\varepsilon}, \cdots, p_n \pm \frac{1}{n\varepsilon})$ differ by at most $\frac{1}{\varepsilon}$ records in expectation.
> > Thus, under an $\varepsilon$-DP training algorithm, with constant probability, any adversary cannot distinguish between datasets sampled with $(p_1, \cdots, p_n)$ and $(p_1 \pm \frac{1}{n\varepsilon}, \cdots, p_n \pm \frac{1}{n\varepsilon})$. As a result, this introduces an inevitable MAE of $\frac{1}{\varepsilon n}$ when estimating the dataset usage $\frac{1}{n}\sum_i p_i$ under an $\varepsilon$-DP training algorithm. Thus, DUCI shows potential for use in DP auditing. However, it remains an interesting open question regarding the connnections between DP auditing via DUCI versus prototypical DP auditing experiment (e.g., via repeated retraining runs).
> >
> > *[1] Hardt, M., & Talwar, K. (2010, June). On the geometry of differential privacy.*

---

> > > ### Comment · Reviewer_H5r7 · 2024-11-27
> > >
> > > Thank you for addressing my questions. It would be great to add the discussion of (1) into the paper. I have updated my score.

---

> > > > ### Author Response · Authors · 2024-11-28
> > > >
> > > > Thank you for your support! We will definitely include the valuable discussion of (1) into our paper. Once again, we deeply appreciate the thoughtful questions you have raised and the time you have dedicated to reviewing both our paper and the rebuttal.

---

### Official Review · Reviewer_THsi · 2024-11-04

**Soundness:** 3
**Presentation:** 3
**Contribution:** 3
**Rating:** 8
**Confidence:** 3

**Summary:**

In this manuscript the authors identify key issues of current techniques that aim to ascertain if a dataset was used to train a Machine Learning model. To alleviate these problems:

1. The authors formally define the concept of Data Usage Cardinality Inference (DUCI). The authors state that, compared to other binary types of inference, DUCI better reflects real world scenarios, where models are trained on fractions of different datasets.
2. The authors propose a way of de-biasing current models that estimate individual membership, i.e. if one individual sample was part of the training dataset. Then, they propose to use these unbiased estimators to compute the overall proportion of the dataset used for training. They also present an asymptotic method to design a confidence interval for this overall proportion used for training.

Finally, the authors provide some numerical experiments where they compare the proposed procedure with four other adapted techniques that also perform DUCI. Throughout their experiments, the authors' de-biased method outperforms the other four techniques.

**Strengths:**

Identifying the dataset used to train a Machine Learning model could have a direct impact on privacy rights or copyright infringement, as mentioned in the Introduction. Hence, I think that this article deals with a relevant problem. I also appreciate that their proposed procedure is cost-effective and intuitive. In my opinion, there is a lot of merit in noticing that Member Identification methods suffer from biases and then presenting an straight-forward tool to address this issue.

**Weaknesses:**

I think that the main weakness here is the presentation. A lot of times the authors describe mathematical objects by vaguely saying what they are or make rushed arguments. However, this approach is not intuitive enough to give any insight about the matter nor formal enough to have any actual meaning. This overshadows the interesting contributions made in this paper.

From Lines 108-115, I wonder what is "a number of population data", what is $\theta(x)_y$ (as this is the first time they use this notation with y as a subscript; in fact, what is y?). What is the reference model modelling, i.e. are these models for membership inference or are these models that represent a real world classifier or regressor?. In Line 285-286, it is difficult to understand what "the probability of observing that i-th record’s likelihood to be a member is greater than randomly sampled population data points" means. This is not even relevant to understand the paper main contributions, so it should be remove it the authors are not willing to explain it clearly or should be rewritten, if they prefer to do so.

Dependence/correlation of records is handled in a confusing manner. In particular, the authors pose the question "Will the ignorance of “correlations” between records make our method sub-optimal?" in Lines 490-491. The answer here is clearly "Yes", as the authors themselves have stated in Lines 444-448 that under special sampling one should divide the dataset into subgroups and then de-bias using the TPR and FPR within each subgroup. However, this additional step, which accounts for possible high-correlation, is not carefully mentioned in Section 4, so I would not assume that this is a fundamental part of their proposed technique. However, it reads as if Lines 489-497 argue that correlation between records is not an issue and that there seems to be limited potential to improve their method in this regard. Maybe the authors here are considering different methods of sampling or different settings but this is not clearly stated in Lines 489-497. This is something that should be addressed.

Regarding the numerical experiments, there are two things to consider: two methods were adapted from Individual Membership Attacks and the other two baseline estimators were inspired by maximum likelihood estimation but rely on additional modeling decisions, like assuming some joint/mean logits follow a normal distribution. Although this choice is based on a theoretical result, as mentioned in Line 298, it is not clear to me that this assumption would not hinder the performance of these baselines. The authors do mention in Line 276 that they use MIA Guess and MIA Score "To demonstrate the importance of debiasing [...]", but I think they would need to address why the MLE with joint logits is presented as an idealized baseline. At least the MLE with average logits has good performance in various experiments, so there is evidence in favor of presenting it as an idealized baseline. However, I feel like the experiments in this paper do indicate that the proposed method performs well, under the scenarios considered here.

**Questions:**

What is the the sampling error mentioned in Line 228 in this particular setting?.
What is the definition of weak independence in Line 269?.

---

> ### Author Response · Authors · 2024-11-22
> **To Reviewer THsi**
>
> We sincerely thank the reviewer for their valuable comments, which have helped improve the clarity of our work. We have addressed all feedback in the [revised paper](https://openreview.net/pdf?id=EUSkm2sVJ6), with the major revisions including:
>
> > Improved clarity of Section 2.1
>
> a. Added definition of $\theta(x)_y$ in Line 98
>
> b. Made the definition of reference model, specific number of population data and how they are used, computation details of RMIA clear in Lines 108-115
>
> c. Removed the expression that could potentially cause confusion in Lines 285–286.
>
> > Dependence/correlation of records is handled in a confusing manner.
>
> Thank you for pointing out the potential confusion. We have improved the clarity of Lines 489-497 in the revised version to separate these two different "correlation" terms:
>
> 1. **Correlation in Lines 444–448 or Footnote 1**:
>    The term $\frac{\text{Corr}_i(\text{TPR}_i - \text{FPR}_i, p_i)}{\text{TPR} - \text{FPR}}$ refers to the "correlation" between the **value** of ground-truth sampling probability $p_i$ of record $i$ and the value of $\text{TPR}_i - \text{FPR}_i$. Here, we (slightly abusively) use the term "correlation" instead of "covariance" because neither $p_i$ nor $\text{TPR}_i - \text{FPR}_i$ are random variables. Note that, although the value of $p_i$ may have a correlation with $\text{TPR}_i - \text{FPR}_i$ for each record $i$, in the DUCI pipeline, the $p_i$ is always a fixed constant, and each record $i$ is Bernoulli-sampled according to $p_i$. A more detailed explanation of this has been added to Appendix D.
>
> 2. **Correlation in Lines 490–491**:
>    This refers to the correlation between the membership probability predictions $\hat{p}_i$ and $\hat{p}_j$ for different records $i$ and $j$ in the dataset. Specifically, "close-to-zero correlation" here means the membership prediction for one record $i$ does not affect the prediction for another record $j$, i.e., $\mathbb{E}[\hat{p}_i \hat{p}_j] = \mathbb{E}[\hat{p}_i]\mathbb{E}[\hat{p}_j]$ for any $i, j \in [|X|]$.
>
> We hope these clarifications resolve the confusion.
>
> > 3. Why assuming some joint/mean logits follow a normal distribution? Why the MLE with joint logits is presented as an idealized baseline?
>
> The motivation and rationale behind the MLE baseline design are threefold. Below, we explain them and will make then clearer in our revision.
>
> **Rationale for using MLE**
> The likelihood ratio test is theoretically proven to be optimal for binary hypothesis testing by Neyman-Pearson lemma [1]. Given that DUCI can naturally be framed as a multi-hypothesis testing problem when the granularity of dataset usage inference is known (we can assume the idealized baseline has access to the granularity of dataset usage, providing it with more information than our method), MLE serves as a natural extension of the pairwise likelihood ratio test to address the DUCI problem.
>
> **Rationale for assuming joint/mean logits follow normal distributions**
> Approximating logits as Gaussian distributions to compute likelihoods is a widely adopted and effective practice in membership inference literature (even in LiRA [2]). This practice is supported by both empirical evidence and theoretical applications, which show that logits often exhibit Gaussian-like behavior across diverse model architectures, data types, and tasks [2, 3, 4]. Therefore, we adopted this practice to compute likelihoods to construct a reasonable and robust baseline that aligns with proven methodologies. To the best of our knowledge, no alternative approaches currently provide apparently better performance or comparable simplicity.
>
> **Necessaty of considering MLE with joint logits**
> While Avg-Logit MLE perform well empirically, averaged statistics inherently introduce information loss. Therefore, it is necessary to include the Joint-Logits MLE as a performance baseline for a lossless scenario. In an idealized setting, where the ground-truth distribution of logits is known, using the joint logits is the most informative choice. However, its empirical sub-optimal performance can be attributed to the curse of dimensionality, i.e., high-dimensional observations (joint logits) typically exhibit a lower signal-to-noise ratio compared to one-dimensional observations (averaged logits).
>
> [1] Neyman, J., & Pearson, E. S. (1933). IX. On the problem of the most efficient tests of statistical hypotheses.
>
> [2] Carlini, N., Chien, S., Nasr, M., Song, S., Terzis, A., & Tramer, F. (2022, May). Membership inference attacks from first principles.
>
> [3] Top-nσ: Not All Logits Are You Need, Chenxia et al, 2024
>
> [4] Lee, J., Xiao, L., Schoenholz, S., Bahri, Y., Novak, R., Sohl-Dickstein, J., & Pennington, J. (2019). Wide neural networks of any depth evolve as linear models under gradient descent.

---

> > ### Author Response · Authors · 2024-11-22
> >
> > > 4. What is the the sampling error mentioned in Line 228 in this particular setting?. What is the definition of weak independence in Line 269?.
> >
> > The sampling error mentioned in Line 228 refers to the standard deviation of the empirical estimates of $P(\hat{m} = 1 \mid m = 0)$ and $P(\hat{m} = 1 \mid m = 1)$ under dataset-level debiasing, over the randomness of $\theta_j$ and the MIA algorithm. We have clarified this meaning in Line 228. Additionally, we have replaced the term "weakly independent" with "approximately independent" to avoid confusion. Line 269 states that we empirically observe near-zero covariance between membership probability predictions $\hat{p}_i$ and $\hat{p}_j$ for any $i, j \in [|X|]$.

---

> > ### Comment · Reviewer_THsi · 2024-11-26
> >
> > Thanks for answering my questions and addressing my suggestions.
> > I've updated my score.

---

> > > ### Author Response · Authors · 2024-11-28
> > >
> > > Thank you again for your detailed and valuable comments, which have been very useful in  improving the quality of our paper. We greatly appreciate your time and effort in reviewing our work!

---

### Official Review · Reviewer_mJ1c · 2024-11-05

**Soundness:** 3
**Presentation:** 4
**Contribution:** 4
**Rating:** 8
**Confidence:** 4

**Summary:**

The paper formalizes the problem of dataset cardinality inference, which aims to measure _how much_ of a given dataset has been used in model training. The paper shows how existing out-of-the-box membership inference methods fail to solve this problem and show how that can be remedied with de-biasing. Experimental results show the benefits of the proposed approach.

**Strengths:**

This paper introduces on a very important problem and gives some solid baselines to tackle it.

The method, error metrics, experimental settings, baselines, and evaluations are thoughtfully designed (in general). For example, I particularly appreciate:
- the extra mile effort in deriving confidence intervals;
- the use of dataset selection methods in experimental evaluations;
- an analysis of why the confidence intervals are large around $p=1/2$;
- the experimental setting of book copyright infringement;

**Weaknesses:**

The main drawback, in my opinion, is that there are approximations involved in deriving the confidence estimates, making them potentially incorrect. There appear to be two approximations (please correct me if I'm mistaken):
1. replacing $TPR_i$, $FPR_i$ with a single TPR/FPR across all samples, so the de-biasing is not exact;
2. assuming independence of $\hat p_i$'s to compute the confidence intervals.

While the authors empirically show in Fig 4 that the correlations in item 2 above are small, it would be nice to see that the bias induced by item 1 is also not too large.

Finally, I would have liked to see some approaches for rigorously correct (asymptotic or non-asymptotic) confidence intervals in addition to the heuristic ones used here. I believe that the XBern confidence intervals given by [Pillutla et al](https://arxiv.org/abs/2305.18447) can be used (XBern confidence intervals for $TPR_i$ and $FPR_i$ can automatically adapt to the correlation, leading to better intervals for $\hat p$).

**Other comments**:
- I do not understand the derivation of footnote 1. It would be nice to expand on it (possibly in the supplement).
- Figure 2 can be clearer if the x axis is in log scale
- Missing relevant refs: [Kandpal et al](https://arxiv.org/pdf/2310.09266) for membership inference of users (groups of data) and is related to dataset inference, [Vyas et al](https://arxiv.org/pdf/2302.10870) for copyright protection, [Zhang et al.](https://arxiv.org/pdf/2406.15968) for a recent MIA

**Questions:**

- **Poor results around $p=0$**: The results of Table 4 show that the method is not very reliable around $p=0$. This would make it unsuitable to answer the question of _if_ a dataset has been used. Are any modifications possible to adapt the proposed method to [dataset inference](https://arxiv.org/abs/2406.06443)?

- Further, how does the proposed method work if our goal is to provide a multiplicative guarantee of the form that $\hat p / p \in (1/c, c)$? These would be more realistic in the small $p$ regime.

- Like differential privacy is designed to protect against membership inference, are there any provable protections against DUCI?
- Why do you think MIA Guess fails to work?

---

> ### Author Response · Authors · 2024-11-22
> **To Reviwer mJ1c**
>
> Thank you for your thoughtful comments and interesting questions! We greatly appreciate their quality and have provided detailed answers to each one below.
>
> > 1. Analysis on the approximation error in the debiasing process and the improved clarity of the derivation in Footnote 1
>
> Our analysis shows that **the simplification in the debiasing process will not introduce errors in many practical sampling scenarios, such as uniform or i.i.d. sampling (which are the common scenario considered in the long line of prior works in binary dataset inference literature listed in the Introduction section). Only in special cases where there is a strong correlation between the probability of sampling $i$-th point $p_i$ and its $TPR_i - FPR_i$, there would be an error. However, this can be effectively mitigated by subgroup debiasing, as shown in Table 2.** We next present the detailed analysis (which is also what footnote 1 analyzed), and we have added details of footnote 1 in the Appendix D.
>
> In Equation (7) and (8), we leverage dataset-level $\text{TPR}$ and $\text{FPR}$ (i.e., $\text{TPR} = \frac{1}{|X|} \sum_i \text{TPR}_i$ and $\text{FPR} = \frac{1}{|X|} \sum_i \text{FPR}_i$) to replace the individual $\text{TPR}_i$ and $\text{FPR}_i$. This simplification avoids the computationally cost (or large sampling errors) of debiasing each $\hat{p}_i$. This simplification is justified because the proportion $p$ is a dataset-level statistic, as analyzed below. To avoid confusion, we introduce $\tilde{p}$ and $\tilde{p}_i$ as the estimators under dataset-level debiasing, and we next prove $\tilde{p} = \frac{1}{|X|} \sum_i \tilde{p}_i = \frac{1}{|X|} \sum_i \frac{\hat{m}_i - \text{FPR}}{\text{TPR} - \text{FPR}}$ is an unbiased estimator of $p$ whenever a correlation term between $\text{TPR}_i - \text{FPR}_i$ and $p_i$ is zero: Given
> \begin{align}
>     \mathbb{E}[\tilde{p}] = \mathbb{E}\left[\frac{1}{|X|} \sum_i \tilde{p}_i\right] = \frac{1}{|X|} \sum_i \mathbb{E}[\tilde{p}_i] = \frac{1}{|X|}\sum_i \mathbb{E}\left[\frac{\hat{m}_i - \text{FPR}}{\text{TPR} - \text{FPR}}\right]
> \end{align}
> Plugging Equation (5) into the above equation, we can get:
> \begin{align}
>     \mathbb{E}[\tilde{p}] & = \frac{1}{\text{TPR} - \text{FPR}} \cdot \frac{1}{|X|} \sum_i \left(p_i \cdot \text{TPR}_i + (1 - p_i) \cdot \text{FPR}_i - \text{FPR}\right) \\
>     & = \frac{\frac{1}{|X|} \sum_i \left[p_i \cdot (\text{TPR}_i - \text{FPR}_i)\right]}{\frac{1}{|X|} \sum_i \left[\text{TPR}_i - \text{FPR}_i\right]}
> \end{align}
> Given $p = \frac{1}{|X|}\sum_i p_i$, note that
> \begin{align*}
>     \frac{1}{|X|} \sum_i \left[p_i \cdot (\text{TPR}_i - \text{FPR}_i)\right] = \frac{1}{|X|} \sum_i p_i \cdot \frac{1}{|X|} \sum_i (\text{TPR}_i - \text{FPR}_i) + \text{Corr}_i(\text{TPR}_i - \text{FPR}_i, p_i).
> \end{align*}
> (Here, we use the term "correlation" instead of "covariance" because $\text{TPR}_i - \text{FPR}_i$ and $p_i$ are not random variables.) Thus, we have:
> \begin{align}
>     \mathbb{E}[\tilde{p}] = p + \frac{\text{Corr}_i(\text{TPR}_i - \text{FPR}_i, p_i)}{\text{TPR} - \text{FPR}}.
> \end{align}
> The correlation term suggests that for many practical sampling methods (e.g., uniform sampling, i.i.d. sampling), this simplification results in an unbiased estimator for $p$ because the correlation is 0. For specialized sampling methods, subgroup debiasing can ensure (empirical) unbiasedness, as discussed in Table 2, by making $\text{TPR}_i - \text{FPR}_i$ constant within each subgroup. This ensures that the correlation term for each subgroup is 0, providing a group-level debiasing approach. Note that the term "correlation" (slightly abused here) is used in the context of how the value of $p_i$, a pre-fixed constant in the DUCI pipeline, is determined. This is distinct from the correlation between membership predictions in Figure 4.
>
> > Figure 2 can be clearer if the x axis is in log scale; Missing relevant refs: Kandpal et al for membership inference of users (groups of data) and is related to dataset inference, Vyas et al for copyright protection, Zhang et al. for a recent MIA
>
> Thanks for the suggestions. We have updated in [revised version](https://openreview.net/pdf?id=EUSkm2sVJ6).
>
> >

---

> > ### Author Response · Authors · 2024-11-22
> >
> > > 2. Levearging the (asymptotic) XBern confidence intervals given by Pillutla et al to derive a confidence interval applicable to correlation.
> >
> > We believe that deriving a tight confidence interval (CI) without assuming independence among membership predictions $\hat{p}_i$ would definitely be informative. However, reliably estimating correlations under DUCI is challenging. ***In DUCI, our goal is to provide a CI for the proportion prediction of a given target model, rather than for the training algorithm used to train that model. This represents the key difference—and the source of difficulty in estimating correlations—between our setting and multiple-tests privacy auditing. Specifically, the correlation $\mathbb{E}[x_1x_2] - \mathbb{E}[x_1]\mathbb{E}[x_2]$ in [1] is typically measured over the randomness of the training algorithm (i.e., empirically calculated across $N$ models trained with the same algorithm). In our case, however, $n = 1$, as each trial involves only a single target model, making it practically infeasible to capture such correlations.***
> >
> > **Theoretical Applicability of XBern CI**
> >
> > The XBern CI is theoretically applicable to DUCI if a randomization algorithm $\mathcal{A}$ is applied to the membership prediction vector $\hat{\mathbf{m}} = [m_i]_{i=1}^{|X|}$. This shuffled membership prediction vector follows the XBern distribution because the shuffling process ensures that each element in the vector becomes an exchangeable Bernoulli sample. Furthermore, the Wilson condition used in the asymptotic XBern CI derivation is a more relaxed condition than the Lyapunov condition in our paper (which we prove to hold in Appendix E.2). Therefore, theoretically, the XBern CI is computable in our setting.
> >
> >
> > **Empirical Challenges with XBern CI**
> >
> > Empirically, however, the XBern CI produces overly loose and misleading results in our context due to the above-mentioned reasons. For instance, as shown in [Figure 1](https://drive.google.com/file/d/1Sp47vWqkurs16ZbzsxiQpLFyBIIvf53x/view?usp=sharing), when we derive a 95\% confidence interval using Proposition 11 from [1], the CI is so loose that all ground-truth $p$ values fall within the interval with probability = 1. More concerningly, the generated CI often has a length greater than 1, which is meaningless since the proportion $p$ is bounded in $[0,1]$. For a sanity check of the XBern CI implementation, we tested its performance by comparing it to our Lyapunov-based CI under settings where they are theoretically equivalent (i.e., using $n = |X|$ duplicated models so that correlation = 0). We observed from [Figure 2](https://drive.google.com/file/d/1efQVcqN9l8WBfR1bs8x0a4GFWcL5Taec/view?usp=sharing) that the First-Order CI produced by XBern was almost identical to our Lyapunov-based CI.
> >
> > We believe that incorporating statistical guarantees into DUCI while accounting for correlations between points in one run would be an interesting direction for future work. However, as also noted in [1], under natural datasets (not specifically crafted to induce correlations), the correlation between points tends to be small. Empirically, our CI is tight and informative: (1) ground-truth $p$ values fall within the CI with around 95\% probability, and (2) its length, along with the MAE (standard deviation), reflects a concentrated Gaussian distribution, indicating that the predicted values are closely aligned with the ground-truth $p$.
> >
> > [1] Pillutla, K., Andrew, G., Kairouz, P., McMahan, H. B., Oprea, A., & Oh, S. (2024). Unleashing the power of randomization in auditing differentially private ml.

---

> > > ### Author Response · Authors · 2024-11-22
> > >
> > > > Poor results around p=0 in Table 4
> > >
> > > This is a really important observation! In Table 4 (the book copyright scenario), all methods exhibit larger absolute errors when $ p = 0 $ with our method showing an error of around 0.1, compared to MIA Score with an error of 0.5. However, **this is not a sign that our method is unsuitable for answering the question of whether a dataset has been used. On the contrary, it highlights the practical motivation for dataset cardinality inference (i.e., different use cases require different thresholds, as specified in the U.S. Copyright Act).**
> > >
> > > As the error decreases with increasing $ p $, and when $ p = 1 $, the error of our method is nearly zero (i.e., MAE = 0.01), even the worst baseline's error reduces to 0.175. This demonstrates that our method performs well when the dataset is used, with only minor confusion when the dataset is not used. **This trend is not observed in the image dataset** as shown in the table below. We believe the explanation for this behavior lies in the nature of language data: **frequently used sentences or phrases with high similarity appear across books, making it nearly impossible for two books to have no phrase-level overlap.** This is why, under the law, a low fraction of similarity in certain content types—such as axioms or public knowledge—is considered fair use.
> > >
> > > **Table:** *(Image Data)* Mean Absolute Error (MAE) $\mathbb{E}[|\hat{p}_i - p|]$ for all methods under different proportions $p$.
> > > | $p$   | MIA Guess | MIA Score | Our Method     |
> > > |---------|-----------|-----------|----------------|
> > > | 0.0     | 0.3025    | 0.2925    | **0.0208**     |
> > > | 0.2     | 0.2375    | 0.1788    | **0.0214**     |
> > > | 0.4     | 0.1635    | 0.0624    | **0.0261**     |
> > > | 0.6     | 0.0937    | 0.0535    | **0.0223**     |
> > > | 0.8     | 0.0407    | 0.1732    | **0.0165**     |
> > > | 1.0     | 0.0526    | 0.2905    | **0.0135**     |
> > > |---------|-----------|-----------|----------------|
> > > |$\max_p \text{MAE}$ | 0.3025    | 0.2925    | **0.0261**     |

---

> ### Author Response · Authors · 2024-11-22
>
> ***Regarding the Concern About the Performance of Our Method in Determining Dataset Usage***
>
> As discussed in Lines 35–46 and shown in Figure 1, while methods restricted to binary predictions under an all-or-none dataset usage scenario cannot ensure consistent predictions for partial utilization, **a method providing fine-grained estimates can naturally be reduced to solve the binary problem.**
>
> To illustrate this, consider the null hypothesis ($H_0: s = s' + \tau$, i.e.,  the target model is not trained on the protected dataset) used in prior binary dataset inference literature. For different contexts, $s$ and $s'$ can take the following forms:
> 1. **Dataset Inference [1]**: $s$ and $s'$ represent the distances to the decision boundary measured on a private dataset and a population dataset, respectively. This hypothesis assumes that if the model was trained on the private dataset, the distance measured on the private dataset would exceed that on the public dataset.
> 2. **LLM Dataset Inference [2]**: $s$ and $s'$ are the weighted aggregations of 52 MIA scores over the private and population datasets. This hypothesis assumes that merged MIA scores would be significantly higher for members than non-members over enough samples.
> 3. **Backdoor Watermarks [3]**: $s$ and $s'$ are the confidence score on the target label given backdoored inputs and given clean inputs. This hypothesis assumes that a model trained on a poisoned dataset (if successfully backdoored) will assign higher confidence to the target class when triggered, but not for clean inputs.
>
> For DUCI, a straightforward simplification to the dataset inference problem can be made by set $s = \hat{p}$, $s' = 0$, and $\tau$ serves as a threshold, which may vary depending on the data type. Below, we report the performance of our method adapted for solving the binary dataset inference problem.
>
> **Table:** Comparison of p-values between DUCI and binary dataset usage algorithms for determining whether a dataset $X$ (size 500) has been used. The complete training dataset of the target model has a size of 25,000. For p-values, a smaller value for **Dataset Used** is better, while a larger value for **Dataset Not Used** is better.
>
> | **Methods**                           | **p-value (Dataset Used ↓)** | **p-value (Dataset Not Used ↑)** |
> |---------------------------------------|------------------------------|-----------------------------------|
> | Backdoor Watermark (poison 30% of $X$) | $7.10 \times 10^{-5}$      | 0.334                            |
> | Backdoor Watermark (poison 100% of $X$) | $\mathbf{6.18 \times 10^{-54}}$ | **1.000**                        |
> | Dataset Inference                     | $7.27 \times 10^{-10}$     | 0.937                            |
> | Ours                                  | $\mathbf{3.15 \times 10^{-51}}$ | **1.000**                        |
>
> Consistent with the performance shown in Figure 1, all methods can perfectly solve the binary dataset usage problem when the significance level is set to common thresholds such as 0.05 or 0.01. **For backdoor watermark methods, the main challenge lies in the successful injection of backdoor when the dataset is not fully sampled or when the protected dataset's relative size is small. This will significantly impact performance, e.g., poisoning even 30% of X performs poorly when $|X|$ is small.
> Our method performs exceptionally well, achieving comparable results to backdoor watermarking when the entire dataset is poisoned.** In principle, the performance of Dataset Inference should be close to our method; however, the slight drop in performance may be attributed to the choice of signal, as the loss-based score is less distinguishable in distribution than the likelihood ratio-based score. We did not compare with [2] as combining multiple MIAs is orthogonal to our approach. Our method can debias any number of MIAs using the same reference models without retraining, with combination possible after debiasing if needed.
>
> Finally, it is important to note that directly comparing the reported error of DUCI at $p = 0$ and $p = 1$ to that of dataset inference is inherently unfair. DUCI predicts a continuous value, whereas dataset inference is a simple binary classification task.
>
> *[1] Maini, P., Yaghini, M., & Papernot, N. (2021). Dataset inference: Ownership resolution in machine learning.*
>
> *[2] Maini, P., Jia, H., Papernot, N., & Dziedzic, A. (2024). LLM Dataset Inference: Did you train on my dataset?*
>
> *[3] Li, Y., Zhu, M., Yang, X., Jiang, Y., Wei, T., & Xia, S. T. (2023). Black-box dataset ownership verification via backdoor watermarking.*

---

> ### Author Response · Authors · 2024-11-22
>
> > 6. Further, how does the proposed method work if our goal is to provide a multiplicative guarantee of the form that $\hat{p}/p \in (1/c, c)$? These would be more realistic in the small regime.
>
> This is an important and relevant question. However, according to the standard packing argument [1], providing a multiplicative guarantee for DUCI (regardless of the method used) is fundamentally impossible. A straightforward example to think about this is: in the extreme case where the ground truth $p = 0$, $\hat{p}$ must also equal 0 to keep the multiplicative ratio bounded. Below, we provide a more detailed explanation.
>
> Let the ground truth membership probability for record $i$ in the training dataset be $p_i$. By definition, a sampled dataset generated under sampling probabilities $(p_1, \ldots, p_n)$ and $(p_1 \pm \frac{1}{10n}, \ldots, p_n \pm \frac{1}{10n})$ can be identical with at least constant probability $\frac{9}{10}$ based on the union bound. As a result, with constant probability, a DUCI algorithm cannot reliably distinguish between datasets sampled with $(p_1, \ldots, p_n)$ versus $(p_1 \pm \frac{1}{n}, \ldots, p_n \pm \frac{1}{n})$, leading to an unavoidable additive error of $\frac{1}{n}$ on either $(p_1, \ldots, p_n)$ or $(p_1 \pm \frac{1}{n}, \ldots, p_n \pm \frac{1}{n})$. Consequently, for any fixed $c \geq 1$, as $p_i \to 0$, the multiplicative error $\frac{\hat{p}_i}{p_i}$ either grows to infinity or shrinks to zero, falling outside the range of $(1/c, c)$.
>
> However, as discussed in Lines 408–410, additive error is a more consistent and meaningful metric for the DUCI problem. This is because DUCI is fundamentally a discrete counting problem, where the focus is on the number of incorrect counts, making additive error a more appropriate measure. For instance, in a small protected dataset of size 10, the unit of the additive error rate is 0.1, which consistently corresponds to a single misprediction. In contrast, using (relative) multiplicative error will lead to nonsensical results: a single misprediction when $p = 0.1$ produces the same ratio as mispredicting all 10 points when $p = 1.0$, which is clearly unreasonable.
>
> > 7. Like differential privacy is designed to protect against membership inference, are there any provable protections against DUCI?
>
> Indeed, when the training algorithm satisfies differential privacy, it is possible to prove upper bounds for the error of DUCI (still via standard packing argument [1]). Loosely speaking (as our goal here is not to derive a tight bound for DUCI), given the ground truth membership probability be $p_i$ for record $i$ in the training dataset, the sampled dataset under sampling probability $(p_1, \cdots, p_{n})$ and $(p_1\pm\frac{1}{n\varepsilon}, \cdots, p_n\pm\frac{1}{n\varepsilon})$ only differ by at most $\frac{1}{\varepsilon}$ records in expectation by definition. Thus under $\varepsilon$-DP training algorithm, with constant probability any adversary could not distinguish between the datasets sampled by $(p_1, \cdots, p_n)$ and $(p_1\pm\frac{1}{\varepsilon n}, \cdots, p_n\pm \frac{1}{\varepsilon n})$. This in turn, causes an inevitable MAE of $\frac{1}{\varepsilon n}$ for estimating the dataset usage $\frac{1}{n}\sum_ip_i$ under $\varepsilon$-DP training algorithm. It is an interesting open question regarding the connections between DP auditing via DUCI versus prototypical DP auditing experiment (e.g., via repeated retraining runs).
>
> *[1] Hardt, M., & Talwar, K. (2010, June). On the geometry of differential privacy.*
>
> > 8. Why do you think MIA Guess fails to work?
>
> The high-level reason, as discussed in Section 3.2 *Errors in Optimal Point-Wise Membership Inference*, is that per-point MIA can make errors. These errors may arise from the intrinsic randomness of the algorithm or inability to capture precise membership information from model outputs. Under DUCI, these errors accumulate across the training set, causing the aggregated MIA guess to deviate significantly from the true $p$. The specific reasons of these errors are not fixed as discussed in prior works [2,3].
>
> Some sources of error are intertwined. For instance, when the score used for MIA is challenging to be perfectly normalized across data points, the optimal threshold may vary for different points. In such cases, naive threshold sweeping becomes less effective. Methods like RMIA, which apply thresholding to the rank of score within the population distribution rather than directly thresholding the score, can be more robust in these scenarios.
>
> However, the objective of this work is not to design the best MIA but to debias any given MIA to perform effectively in DUCI. As such, we do not delve into the specific limitations of existing MIAs.
>
> *[2] Aubinais, E., Gassiat, E., & Piantanida, P. (2023). Fundamental Limits of Membership Inference Attacks on Machine Learning Models.*
>
> *[3] Maini, P., et al. (2024). LLM Dataset Inference: Did you train on my dataset?*

---

> > ### Comment · Reviewer_mJ1c · 2024-11-26
> >
> > I appreciate the detailed and careful responses of the authors and the painstaking additional comparisons! I have raised my score accordingly.
> >
> > Some further comments (it is enough for the authors to think about these and address them in the revision; no need to reply here):
> >
> > **Text vs. image data**: I agree that it can be true for some settings but I do not fully buy this argument. For typical language modeling tasks, each "example" is an entire sequence (which can be several 1000s of tokens long). Even though two books can have meaningful phrase-level overlap, I would expect enough differences in such long sequences that it would be possible to notice a difference. While the empirical observation is very interesting, the authors may wish to nuance their argument.
> >
> > **Impossibility of multiplicative guarantees**: This is super interesting, it would be great to add this to the paper somewhere.
> >
> > Thanks again and all the best!

---

> > > ### Author Response · Authors · 2024-11-26
> > >
> > > Thank you so much for your support, your efforts in reviewing our paper and rebuttals, and your willingness to raise the score! We really enjoy the discussion, and all the questions are both interesting and valuable. We will definitely incorporate the impossibility of multiplicative guarantees and further analysis on the different behaviors of text vs. image data into our revision. Especially, regarding the distinguishability of text samples, we will add practical experiments (e.g., with carefully designed different overlap levels between text sequences) to nuance the argument.

---

### Meta-Review · Area_Chair_A8jB · 2024-12-21

**Metareview:**

This submission introduces Dataset Usage Cardinality Inference (DUCI): a framework for modeling and inference of the proportion of data used when training a model. The authors (elegantly) motivate the problem in terms of the US Copyright Act, provide statistical guarantees for DUCI, and illustrate their approach on image and text datasets.

The reviewers agreed that the paper is well-written and the problem is timely. They also raise several issues and questions surrounding notation, which were appropriately addressed in the rebuttal. This is a nice contribution that extends the literature on membership inference attacks in a non-trivial direction.

I encourage the authors to seriously consider the reviewers' comments when preparing the final version of the manuscript, specifically those related to the presentation of the experiments and the theoretical results.

**Additional Comments On Reviewer Discussion:**

The reviewers remained positive about the paper after the rebuttal. The points raised by the reviewers in terms of presentation, experiments, and theory were appropriately addressed in the rebuttal.

---

### Decision · Program_Chairs · 2025-01-22

Accept (Oral)